# Hyperoxidation of mitochondrial peroxiredoxin limits H$_2$O$_2$-induced cell death in yeast

Gaetano Calabrese[1] ID, Esra Peker[1], Prince Saforo Amponsah[2,3], Michaela Nicole Hoehne[1], Trine Riemer[1], Marie Mai[3], Gerd Patrick Bienert[4] ID, Marcel Deponte[5] ID, Bruce Morgan[3,*] ID & Jan Riemer[1,**] ID

## Abstract

Hydrogen peroxide (H$_2$O$_2$) plays important roles in cellular signaling, yet nonetheless is toxic at higher concentrations. Surprisingly, the mechanism(s) of cellular H$_2$O$_2$ toxicity remain poorly understood. Here, we reveal an important role for mitochondrial 1-Cys peroxiredoxin from budding yeast, Prx1, in regulating H$_2$O$_2$-induced cell death. We show that Prx1 efficiently transfers oxidative equivalents from H$_2$O$_2$ to the mitochondrial glutathione pool. Deletion of *PRX1* abrogates glutathione oxidation and leads to a cytosolic adaptive response involving upregulation of the catalase, Ctt1. Both of these effects contribute to improved cell viability following an acute H$_2$O$_2$ challenge. By replacing *PRX1* with natural and engineered peroxiredoxin variants, we could predictably induce widely differing matrix glutathione responses to H$_2$O$_2$. Therefore, we demonstrated a key role for matrix glutathione oxidation in driving H$_2$O$_2$-induced cell death. Finally, we reveal that hyperoxidation of Prx1 serves as a switch-off mechanism to limit oxidation of matrix glutathione at high H$_2$O$_2$ concentrations. This enables yeast cells to strike a fine balance between H$_2$O$_2$ removal and limitation of matrix glutathione oxidation.

**Keywords** cell death; hydrogen peroxide; hyperoxidation; mitochondria; peroxiredoxin

**Subject Categories** Autophagy & Cell Death; Membranes & Trafficking

The EMBO Journal (2019) 38: e101552

## Introduction

Reactive oxygen species (ROS) are an unavoidable consequence of life in an oxygen-rich environment. Once considered solely as harmful molecules, which cells seek to remove as efficiently as possible, it is now accepted that some ROS have important physiological functions. In this regard, one of the best understood ROS is H$_2$O$_2$, which acts as a second messenger in several key cellular signaling pathways (Sundaresan *et al*, 1995; Delaunay *et al*, 2002; Sobotta *et al*, 2015; Stocker *et al*, 2018). On the other hand, it remains unequivocal that high concentrations of H$_2$O$_2$ are toxic and can lead to cellular dysfunction and cell death. Presumably therefore, cells strive to tightly regulate H$_2$O$_2$ to permit sufficiently large fluctuations in H$_2$O$_2$ concentration for signaling purposes, while simultaneously preventing accumulation of H$_2$O$_2$ to toxic levels.

Surprisingly, the exact mechanism(s) by which H$_2$O$_2$ leads to cell death remain poorly understood (Whittemore *et al*, 1995; Day *et al*, 2012; Uhl *et al*, 2015). The mediocre reactivity of H$_2$O$_2$ with most biological molecules argues against direct oxidation of cellular biomolecules being a principal driver of cell death, at least at physiologically relevant H$_2$O$_2$ concentrations (Winterbourn & Metodiewa, 1999; Stone, 2004; Winterbourn, 2008). Other possible triggers of H$_2$O$_2$-induced cell death have been proposed, including induction of apoptosis (Greetham *et al*, 2013), depletion of reduced cytosolic thioredoxins (Day *et al*, 2012), and disruption of redox signaling pathways (Sies, 2017), but well-defined molecular mechanisms remain largely elusive. Therefore, in this study we sought to address the molecular underpinnings of cellular H$_2$O$_2$ toxicity, using budding yeast as a model system.

We hypothesized that mitochondrial matrix-localized processes may play an important role in H$_2$O$_2$ toxicity, for two reasons. First, the mitochondrial respiratory chain is thought to be a major source of cellular H$_2$O$_2$, and thus, the matrix is in close proximity to key sites of H$_2$O$_2$ production (Murphy, 2009; Quinlan *et al*, 2013). Specifically, the "leakage" of electrons from respiratory chain complexes to molecular oxygen leads to the generation of superoxide anions. Superoxide dismutases located in both the mitochondrial matrix (Sod2) and the cytosol/intermembrane space (Sod1) facilitate the subsequent rapid dismutation of superoxide to H$_2$O$_2$. Second, in comparison with the cytosol, the mitochondrial matrix appears to be poorly equipped with H$_2$O$_2$-detoxifying enzymes. The only enzymes in the yeast mitochondrial matrix that are known to be able to react efficiently with H$_2$O$_2$ are the 1-Cys peroxiredoxin, Prx1, and perhaps the glutathione peroxidase homolog, Gpx2 (Park

1   Department for Chemistry, Institute for Biochemistry, University of Cologne, Cologne, Germany
2   Department for Biology, Cellular Biochemistry, University of Kaiserslautern, Kaiserslautern, Germany
3   Institute of Biochemistry, University of the Saarland, Saarbruecken, Germany
4   Department of Physiology and Cell Biology, Leibniz-Institute of Plant Genetics and Crop Plant Research (IPK), Gatersleben, Germany
5   Department of Chemistry/Biochemistry, University of Kaiserslautern, Kaiserslautern, Germany
    *Corresponding author. Tel: +49-681-302-3339; E-mail: bruce.morgan@uni-saarland.de
    **Corresponding author. Tel: +49-221-470-7306; E-mail: jan.riemer@uni-koeln.de

*et al*, 2000; Ukai *et al*, 2011). Nonetheless, many aspects of $H_2O_2$ handling in the matrix remain poorly understood, for example, the efficiency of the matrix-localized $H_2O_2$ removal systems. Furthermore, the crosstalk between matrix $H_2O_2$ and different matrix redox couples is unclear, and often, conflicting results are present in the literature. For example, the exact reductive mechanisms for both Prx1 and Gpx2 remain a matter of debate, particularly regarding the specific roles of glutathione, glutaredoxins, thioredoxins and thioredoxin reductase (Pedrajas *et al*, 2000, 2010, 2016; Avery & Avery, 2001; Tanaka *et al*, 2005; Greetham & Grant, 2009).

To gain further insight into matrix $H_2O_2$ handling in general and more specifically into possible mechanisms of $H_2O_2$-mediated toxicity, we employed genetically encoded probes that enable subcellular compartment-specific measurement of $H_2O_2$, the glutathione redox potential ($E_{GSH}$), and Prx1 oxidation, together with biochemical assessment of cysteine redox states, transcriptome analyses and cell death assays. We found that the matrix glutathione pool is significantly more sensitive to $H_2O_2$-induced oxidation than the cytosolic glutathione pool. However, we found that $H_2O_2$-induced matrix glutathione oxidation is completely dependent upon the presence of Prx1. Deletion of *PRX1* eliminated $H_2O_2$-induced oxidation of matrix glutathione and elicited a transcriptional response that increased levels of the cytosolic catalase, Ctt1, showing that cells can recognize, and respond to, impaired matrix redox homeostasis. The loss of glutathione oxidation in the matrix and the improved Ctt1-dependent $H_2O_2$-handling capacity of the cytosol synergistically rendered cells more resistant to an acute $H_2O_2$ treatment. We subsequently generated a range of matrix-targeted thiol peroxidases and mutant variants thereof, with differing abilities to transfer oxidative equivalents from $H_2O_2$ to glutathione. By replacing endogenous Prx1 with these peroxiredoxin variants, we revealed a striking correlation between matrix glutathione oxidation and cell death. In wild-type cells, we found that the degree of cell death is limited by hyperoxidation-based inactivation of Prx1 at high $H_2O_2$ levels, which restricts oxidation of the matrix glutathione pool. In summary, Prx1 hyperoxidation allows cells to strike a fine balance between $H_2O_2$ removal and limitation of mitochondrial glutathione oxidation, which is strongly predictive of cell death.

## Results

### Exogenous $H_2O_2$ elicits compartment-specific $E_{GSH}$ and $H_2O_2$ responses

Little is known about the dynamic handling of $H_2O_2$ in the mitochondrial matrix of intact cells. We thus sought to characterize $H_2O_2$ flux inside the matrix and assess the impact of increased $H_2O_2$ on matrix reductive systems, particularly the glutathione pool. To this end, we made use of the genetically encoded fluorescent probes, roGFP2-Tsa2$\Delta C_R$ and Grx1-roGFP2, which allow the real-time monitoring of the basal $H_2O_2$ level and $E_{GSH}$, respectively, in specific subcellular compartments (Gutscher *et al*, 2008; Morgan *et al*, 2016; Fig 1A and B). Both probes comprise a redox-sensitive green fluorescent protein (roGFP2; Hanson *et al*, 2004) genetically fused with a specific redox enzyme. For roGFP2-Tsa2$\Delta C_R$, this is the *Saccharomyces cerevisiae* cytosolic typical 2-Cys peroxiredoxin, Tsa2, from which the resolving cysteine has been removed to increase the

sensitivity of the probe to $H_2O_2$. In the case of Grx1-roGFP2, it is the human glutaredoxin, Grx1. The roGFP2-Tsa2$\Delta C_R$ probe responds directly to $H_2O_2$, with the Tsa2$\Delta C_R$ moiety serving to efficiently transfer oxidative equivalents from $H_2O_2$ to roGFP2. This probe is predominantly reduced by endogenous GSH/glutaredoxins, which directly reduce the roGFP2 moiety. RoGFP2-Tsa2$\Delta C_R$ oxidation is therefore determined by rapid $H_2O_2$-driven oxidation and much slower GSH/glutaredoxin-driven reduction (Morgan *et al*, 2016; Roma *et al*, 2018). Conversely, Grx1-roGFP2 will predominantly only respond indirectly to $H_2O_2$, via $H_2O_2$-induced glutathione disulfide (GSSG) production and the concomitant increase in $E_{GSH}$. The readout of roGFP-based probes is commonly presented as degree of probe oxidation to allow comparison between different experiments (OxD; for calculation, see Materials and Methods and Gutscher *et al*, 2008). OxDs of 0 and 1 therefore indicate fully reduced and fully oxidized roGFP2 moieties, respectively (Appendix Fig S1A; for more detailed discussion of the probe mechanisms, see, e.g., Roma *et al*, 2018).

We targeted each probe to either the cytosol or the mitochondrial matrix and monitored the probe response to the addition of exogenous $H_2O_2$ using a fluorescence plate reader-based system (Appendix Fig S1B, and Fig 1C and D). We observed that the matrix roGFP2-Tsa2$\Delta C_R$ probe exhibited a significantly smaller response to exogenous $H_2O_2$, applied at initial concentrations of 0.1 and 1 mM, than the cytosolic roGFP2-Tsa2$\Delta C_R$ probe (Fig 1C). In both subcellular compartments, the OxD of the roGFP2-Tsa2$\Delta C_R$ probe decreased over time in the absence of $H_2O_2$, an observation explained by the depletion of oxygen in our plate reader-based assay (Morgan *et al*, 2016). In contrast, upon addition of exogenous $H_2O_2$, the matrix-targeted Grx1-roGFP2 probe exhibited a significantly larger response than the cytosol-localized Grx1-roGFP2 (Fig 1D). In summary, matrix $E_{GSH}$ is apparently more sensitive to perturbation by exogenous $H_2O_2$, even though the roGFP2-Tsa2$\Delta C_R$ probe reports less $H_2O_2$ in the matrix than in the cytosol. We therefore sought to identify the cause(s) of these compartment-specific differences.

### Cytosolic enzymes protect the matrix against both exogenous and mitochondria-derived $H_2O_2$

The smaller roGFP2-Tsa2$\Delta C_R$ response in the matrix compared to the cytosol could be explained by either (i) efficient cytosolic scavenging systems as well as the two mitochondrial membranes limiting the amount of $H_2O_2$ that reaches the matrix or (ii) by more efficient scavenging of $H_2O_2$ in the matrix limiting the amount of $H_2O_2$ available to react with the probe.

An initial insight into these two possibilities came from monitoring roGFP2-Tsa2$\Delta C_R$ and Grx1-roGFP2 probe responses to antimycin A treatment. Antimycin A is an inhibitor of respiratory chain complex III that results in release of superoxide anions on the IMS side of the inner mitochondrial membrane. Superoxide anions will be rapidly dismutated, both enzymatically and spontaneously, to $H_2O_2$. Antimycin A treatment led to a larger roGFP2-Tsa2$\Delta C_R$ response in the matrix compared to the cytosol, i.e., the opposite of the situation following treatment with exogenous $H_2O_2$ (Fig 1E). Antimycin A treatment also induced a small deflection of $E_{GSH}$ in both the matrix and the cytosol. The response in the matrix appeared to be slightly larger than in the cytosol, although the overall response was very limited in both compartments (Fig 1F).

Currently, we do not understand why the comparatively strong roGFP2-Tsa2ΔCR probe response upon antimycin A treatment did not result in a respective $E_{GSH}$ deflection. A possible hint might be the antimycin A-specific response dynamics. Compared to addition of external $H_2O_2$, antimycin A induced a comparatively late oxidation of the roGFP2-Tsa2ΔCR probe without recovery. In conclusion, the observation of opposing compartment-specific responses to antimycin A and exogenous $H_2O_2$ treatment indicates that the subcellular localization of $H_2O_2$ production/influx is an important determinant of subcellular compartment-specific $H_2O_2$ levels. Likely, cellular $H_2O_2$ scavenging enzymes significantly limit the (sub)cellular diffusion of $H_2O_2$ leading to the generation of steep intracellular $H_2O_2$ gradients (Winterbourn, 2008; Lim et al, 2015; Travasso et al, 2017).

To further test this hypothesis, we monitored the response of cytosolic and matrix-localized roGFP2-Tsa2ΔCR probes to exogenous peroxide in either wild-type cells or cells deleted for the genes encoding the two cytosolic typical 2-Cys peroxiredoxins, Tsa1 and Tsa2. Tsa1, in particular, is a highly abundant protein and thought to be an important cytosolic scavenger of $H_2O_2$ (Iraqui et al, 2009). In Δtsa1Δtsa2 cells, we saw that cytosolic and matrix roGFP2-Tsa2ΔCR responses (although starting from a different initial steady state) to exogenous $H_2O_2$ were much more similar than in wild-type cells (Fig 1G). These data thus further support the hypothesis that cytosolic $H_2O_2$ scavenging enzymes, including Tsa1 and Tsa2, limit the amount of exogenous $H_2O_2$ that can diffuse through the cytosol to ultimately reach the mitochondrial matrix. Interestingly, we also observed that Tsa1 and Tsa2 are important for the

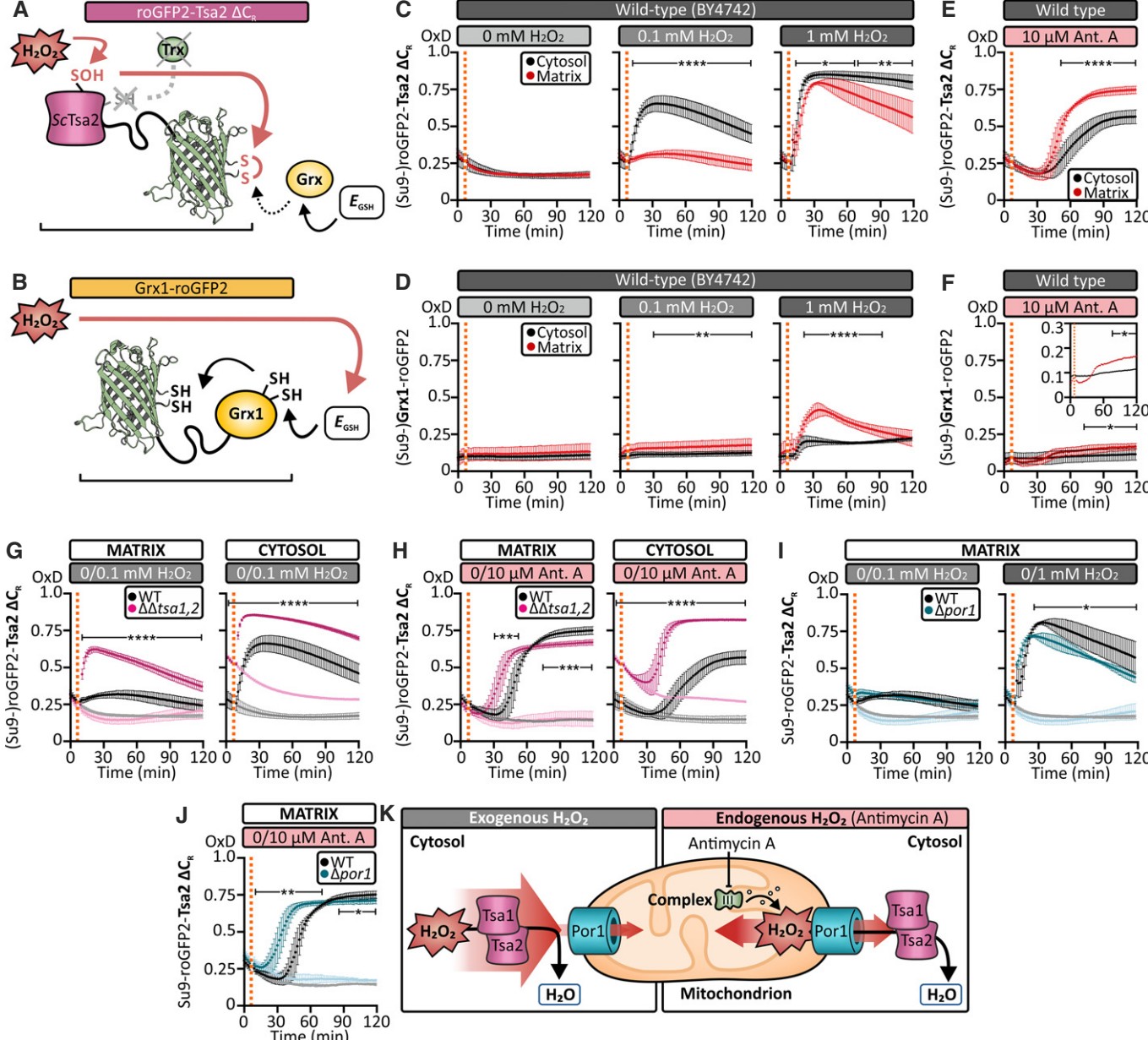

**Figure 1.**

◄

**Figure 1.  Matrix $E_{GSH}$ is more sensitive to $H_2O_2$ than its cytosolic counterpart.**

A    A scheme depicting the $H_2O_2$ sensing mechanism of the peroxiredoxin-based $H_2O_2$ sensor roGFP2-Tsa2ΔC$_R$.

B    A scheme depicting the mechanism of the $E_{GSH}$ sensor, Grx1-roGFP2.

C, D  The response of cytosolic and mitochondrial matrix-localized roGFP2-Tsa2ΔC$_R$ (C) and Grx1-roGFP2 (D) probes to bolus of exogenous $H_2O_2$ at the indicated concentrations. Probes were expressed in wild-type BY4742 yeast cells grown to exponential phase in SGal (−Leu) medium.

E, F  The response of cytosolic and mitochondrial matrix-localized roGFP2-Tsa2ΔC$_R$ (E) and Grx1-roGFP2 (F) probes to 10 μM complex antimycin A (a complex III inhibitor). Probes were expressed in wild-type BY4742 yeast cells grown to exponential phase in SGal (−Leu) medium.

G    The response of mitochondrial matrix-localized (left panel) and cytosolic (right panel) roGFP2-Tsa2ΔC$_R$ probes, expressed in BY4742 wild-type and Δtsa1Δtsa2 cells, to the addition of exogenous $H_2O_2$ at the indicated concentrations. Cells were grown to exponential phase in SGal (−Leu) medium. The lighter colored curves are controls, showing the probe response upon the addition of water.

H    The response of mitochondrial matrix-localized (left panel) and cytosolic (right panel) roGFP2-Tsa2ΔC$_R$ probes, expressed in wild-type and Δtsa1Δtsa2 cells, to the addition of 10 μM antimycin A. Cells were grown to exponential phase in SGal (−Leu) medium. The lighter colored curves are controls showing the probe responses upon addition of 0.1% (v/v) ethanol.

I    The response of a mitochondrial matrix-localized roGFP2-Tsa2ΔC$_R$ probe, expressed in wild-type and Δpor1 cells, to bolus at exogenous $H_2O_2$ at the indicated concentrations. Cells were grown to exponential phase in SGal (−Leu) medium. Lighter colored curves are controls showing the probe response upon the addition of water.

J    The response of a mitochondrial matrix-localized roGFP2-Tsa2ΔC$_R$ probe, expressed in wild-type and Δpor1 cells, to the addition of 10 μM antimycin A. Cells were grown to exponential phase in SGal (−Leu) medium. Lighter colored curves are controls showing the probe response upon addition of 0.1% (v/v) ethanol.

K    Model. Cytosolic peroxiredoxins protect the mitochondrial matrix from internal (mitochondrial) and external $H_2O_2$.

Data information: In all panels, OxD refers to the degree of sensor oxidation. Error bars represent the standard deviation ($n = 3$ biological replicates, with cells obtained from three independent cultures for every strain and probe combinations. For each biological replicate, three technical replicates were performed). Significance was assessed with Student's 2-tailed, unpaired, $t$-test. *$P < 0.05$; **$P < 0.01$; ***$P < 0.001$; and ****$P < 0.0001$.

detoxification of mitochondria-derived $H_2O_2$, as a matrix roGFP2-Tsa2ΔC$_R$ probe in Δtsa1Δtsa2 cells responded more rapidly to antimycin A treatment than in wild-type cells (Fig 1H). Thus, release of $H_2O_2$ to the cytosol likely also constitutes a mitochondrial $H_2O_2$ detoxification pathway.

We next tested whether transfer over the mitochondrial membranes contributes to a decreased roGFP2-Tsa2ΔC$_R$ response in the matrix compared to the cytosol. In other systems, it has been demonstrated that the velocity of $H_2O_2$ transfer over membranes is increased by the presence of specific transporters, e.g., aquaporin 8 in the NADPH oxidase 2-dependent signaling cascade (Bertolotti et al, 2016). The outer membrane of mitochondria (OMM) contains porins (in yeast Por1 and the less expressed Por2) that have been shown to facilitate small molecule transport (Kmita et al, 2004; Kojer et al, 2012). Indeed, we found that the response to 1 mM exogenous $H_2O_2$ of a matrix roGFP2-Tsa2ΔC$_R$ in a Δpor1 strain was decreased compared to a wild-type strain (Fig 1I). Conversely, following antimycin A treatment, the matrix roGFP2-Tsa2ΔC$_R$ probe responded more readily in Δpor1 cells than in wild-type cells, supporting the hypothesis that Por1 deletion decreases $H_2O_2$ transport to the cytosol (Fig 1J). In summary, Por1 appears to facilitate the bi-directional transport of $H_2O_2$ across the OMM. Collectively, our data indicate that cytosolic peroxiredoxins and porins in the OMM contribute a major line of defense for mitochondria from external $H_2O_2$ and support the efficient removal of mitochondria-derived $H_2O_2$ (Fig 1K).

## Ctt1 upregulation constitutes an "adaptive response" in matrix redox enzyme mutants

We next investigated the contribution of matrix $H_2O_2$ scavenging enzymes toward regulation of matrix $H_2O_2$ level. The 1-Cys peroxiredoxin, Prx1, is likely the most important $H_2O_2$ scavenger in the yeast mitochondrial matrix. However, surprisingly, when we deleted this enzyme, the matrix roGFP2-Tsa2ΔC$_R$ response to exogenous $H_2O_2$ was decreased (Fig 2A). This is counterintuitive for an enzyme that has been implicated in efficient matrix $H_2O_2$ handling. Conversely, when we assessed the matrix roGFP2-Tsa2ΔC$_R$ response toward antimycin A treatment, the differences in response between wild-type and Δprx1 cells were lost (Appendix Fig S2A). Collectively, these results suggest that defective matrix $H_2O_2$ handling might result in compensatory responses in the cytosol. To test this hypothesis, we monitored cytosolic $H_2O_2$ handling in strains lacking the matrix redox enzymes Prx1 and Trx3, which have both previously been linked to efficient $H_2O_2$ handling in the matrix (Pedrajas et al, 2000; Greetham & Grant, 2009; Gostimskaya & Grant, 2016). Interestingly, in both deletion strains cytosolic roGFP2-Tsa2ΔC$_R$ responses were attenuated compared to the wild-type cells (Fig 2B and C). Complementation of gene loss by expressing the matrix redox enzymes from a plasmid saw cytosolic roGFP2-Tsa2ΔC$_R$ responses revert to a wild-type-like situation (Fig 2D).

To understand this apparent cytosolic adaptation, we performed RNA-Seq analysis on Δprx1 cells transformed either with an empty p416TEF plasmid (Δprx1 + empty) or with a pEPT-PRX1 plasmid (Δprx1 + Prx1-WT) for the expression of wild-type Prx1 (Fig 2E, and Appendix Fig S2B and C). We found that the transcript encoding the cytosolic catalase, Ctt1, was significantly upregulated, while transcripts encoding other redox enzymes were not significantly enriched above threshold (Fig 2E and Dataset EV1). In line with a role of Ctt1 in the cytosolic "adaptive response", the deletion of CTT1 in a Δprx1 background (Δctt1Δprx1) ablated the decreased response of cytosolic roGFP2-Tsa2ΔC$_R$ that was observed in Δprx1 cells (Fig 2F). In line with the loss of the adaptive response, Δprx1Δctt1 cells exhibit a prolonged lag phase compared to either wild-type, Δprx1, or Δctt1 cells during growth under chronic $H_2O_2$ stress (Appendix Fig S2D). Increased Ctt1 levels appear to be an "add-on" response as deletion of PRX1 in Δtsa1Δtsa2 cells only conferred a small, although significantly, improved cytosolic $H_2O_2$ handling (Appendix Fig S2E). Thus, not only cytosolic redox enzymes protect the matrix from external $H_2O_2$ under "unperturbed" conditions, but also there appears to be a cytosol-based "adaptive response" if matrix redox enzyme systems are impaired.

This Ctt1-dependent, NADPH-independent, response decreases the amount of exogenous $H_2O_2$ that reaches the mitochondrial matrix in cells with compromised matrix $H_2O_2$ detoxification systems (Fig 2G).

## Glutathione reductase activity is limiting in the matrix

The experiments described above indicate that cytosolic redox enzymes significantly limit the amount of exogenous $H_2O_2$ that

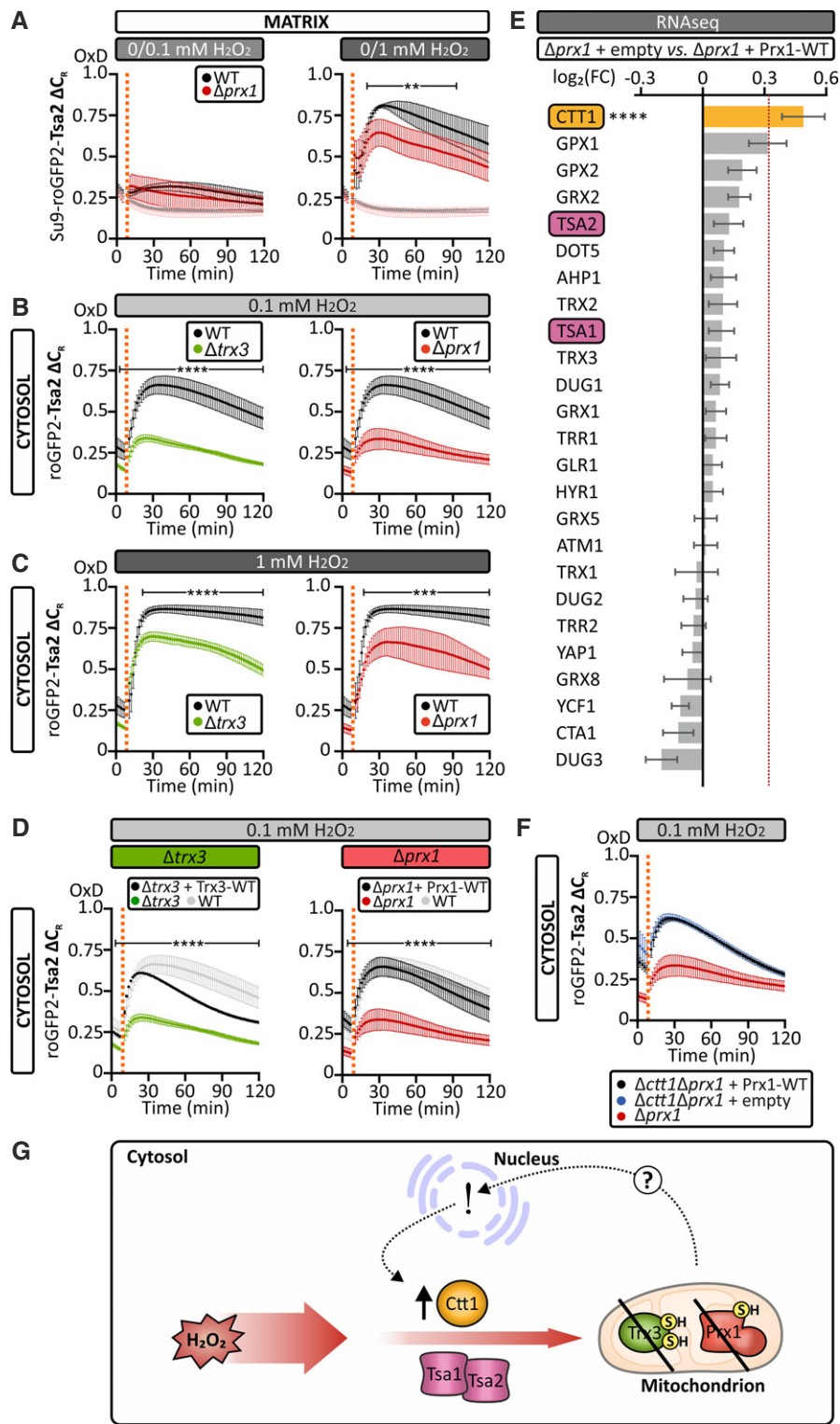

**Figure 2.**

**Figure 2.  Deletion of mitochondrial redox enzymes activates a cytosolic adaptive response.**

A     The response of a mitochondrial matrix-localized roGFP2-Tsa2ΔC$_R$ probe, expressed in wild-type and Δ$prx1$, to the addition of exogenous H$_2$O$_2$ at the indicated concentrations. Cells were grown in SGal (−Leu) medium and harvested at early exponential phase. Lighter colored curves are controls showing the probe response to the addition of water.

B, C  The response of a cytosolic roGFP2-Tsa2ΔC$_R$ probe, expressed in wild-type, Δ$trx3$, and Δ$prx1$ cells, to the addition of 0.1 mM (B) or 1 mM (C) exogenous H$_2$O$_2$. Cells were grown in SGal (−Leu) medium and harvested at early exponential phase.

D     The response of a cytosolic roGFP2-Tsa2ΔC$_R$ probe, in Δ$trx3$ cells transformed with a Trx3 plasmid and Δ$prx1$ cells transformed with a Prx1 plasmid, to the addition of 0.1 mM exogenous H$_2$O$_2$. Cells were grown in SGal medium lacking the appropriate amino acids for plasmid selection and harvested at early exponential phase. The light gray curve in both panels represents the wild-type control, treated with the same concentration of exogenous H$_2$O$_2$.

E     The profile of mRNA expression of the yeast redox or redox-related enzymes in Δ$prx1$ + empty vector cells, compared to Δ$prx1$ + Prx1-WT cells, both grown to an early exponential phase in SGal (−Ura) medium ($n$ = 3 biological replicates, with cells obtained from three independent cultures). Cutoff was set a log$_2$(fold change (FC)) ± 0.32. Raw data are presented in Dataset EV1.

F     The response of a cytosolic roGFP2-Tsa2ΔC$_R$ probe, in Δ$prx1$ cells or Δ$ctt1$Δ$prx1$ cells transformed with either an empty plasmid or a Prx1-WT plasmid to the addition of 0.1 mM exogenous H$_2$O$_2$. Cells were grown in SGal medium lacking the appropriate amino acids for selection and harvested in early exponential phase.

G     Model. Levels of the cytosolic catalase Ctt1 increase when activity of mitochondrial redox enzymes is impaired, leading to an increased cytosolic capacity for H$_2$O$_2$ removal.

Data information: In all panels, OxD refers to the degree of sensor oxidation. Error bars represent the standard deviation ($n$ = 3 biological replicates, with cells obtained from three independent cultures for every strain and probe combinations. For each biological replicate, three technical replicates were performed). Significance was assessed with Student's 2-tailed, unpaired, $t$-test. **$P$ < 0.01; ***$P$ < 0.001; and ****$P$ < 0.0001.

reaches the mitochondrial matrix. Nevertheless, matrix $E_{GSH}$ is still more responsive to exogenous H$_2$O$_2$ than cytosolic $E_{GSH}$ despite the lower concentration of H$_2$O$_2$ that reaches the matrix. We therefore wanted to gain a deeper understanding of the mechanistic basis of this difference. We reasoned that, in the matrix, either GSSG might be less efficiently reduced or H$_2$O$_2$ might more efficiently trigger glutathione oxidation in comparison with the situation in the cytosol. First, we assessed whether glutathione reductase (Glr1) levels are limiting in the matrix. Glr1 is dually localized to the cytosol and matrix. The two Glr1 variants are encoded by one gene and one mRNA that is translated from two different start codons, with the longer form encompassing a matrix-targeting sequence (Outten & Culotta, 2004). Deleting *GLR1* resulted in a higher steady-state Grx1-roGFP2 oxidation in both compartments and a much larger response to exogenous H$_2$O$_2$ compared to wild-type control cells, confirming previous findings (Kojer *et al*, 2012; Morgan *et al*, 2013; Appendix Fig S3A). Notably, complementation of the Δ$glr1$ strain with Glr1 expressed from a plasmid, under the control of a strong constitutive TEF promoter (Δ$glr1$ + Glr1), led to a decreased matrix $E_{GSH}$ response compared to that observed in wild-type cells. However, maintenance of cytosolic $E_{GSH}$ did not benefit to the same extent from Glr1 overexpression. Collectively, these data indicate that Glr1 is limiting for GSSG reduction in the matrix (Appendix Fig S3B). Nonetheless, even with Glr1 overexpression, matrix $E_{GSH}$ still responds more readily to exogenous H$_2$O$_2$ than cytosolic $E_{GSH}$. We thus next asked whether H$_2$O$_2$ also more efficiently elicits GSSG production in the matrix than in the cytosol and if so, what enzymes are important for catalyzing the glutathione-mediated reduction of H$_2$O$_2$.

## Prx1 catalyzes the glutathione-dependent reduction of H$_2$O$_2$ in the matrix

To gain further insight into how H$_2$O$_2$ affects $E_{GSH}$ in the matrix, we monitored the response of matrix $E_{GSH}$ to exogenous H$_2$O$_2$ in Δ$trx3$, Δ$trr2$, and Δ$prx1$ strains, and compared it with wild-type, Δ$tsa1$Δ$tsa2$, and Δ$por1$ cells (Fig 3A). As expected, Δ$tsa1$Δ$tsa2$ cells exhibited an increased response of matrix Grx1-roGFP2 compared to wild-type because more H$_2$O$_2$ reaches the matrix in these cells

(Fig 1G). Conversely, and in line with the roGFP2-Tsa2ΔC$_R$ data (Fig 1I), Δ$por1$ cells exhibited a decreased response of the $E_{GSH}$ sensor. Intriguingly, in the Δ$trx3$, Δ$trr2$, and Δ$prx1$ strains we observed either no $E_{GSH}$ response (Δ$prx1$) or an $E_{GSH}$ response that was greatly decreased compared to wild-type cells (Δ$trx3$, Δ$trr2$), following the addition of exogenous H$_2$O$_2$ at an initial concentration of 1 mM (Fig 3A). To assess whether the cytosolic adaptive response described above (Fig 2) might explain the decreased matrix $E_{GSH}$ response, we applied exogenous H$_2$O$_2$ at the higher initial concentration of 2.5 mM. While the matrix $E_{GSH}$ response in the Δ$trx3$ and Δ$trr2$ strains increased to almost the same level as that observed in wild-type cells, we still observed no matrix $E_{GSH}$ response in Δ$prx1$ cells (Fig 3B). Importantly, at 2.5 mM exogenous H$_2$O$_2$, the matrix roGFP2-Tsa2ΔC$_R$ response in Δ$prx1$ cells was clearly increased compared to the respective response in the wild-type to 1 mM exogenous H$_2$O$_2$ (Fig 3C). This indicates that despite the increased H$_2$O$_2$-handling capacity of the cytosol due to increased Ctt1 levels, Prx1 is crucial in mediating the $E_{GSH}$ response to H$_2$O$_2$. Consistent with the enzymatic activity of Prx1 being required for the H$_2$O$_2$-induced oxidation of glutathione, we observed no $E_{GSH}$ response in Δ$prx1$ cells transformed with a plasmid encoding a Prx1 peroxidatic cysteine mutant, Prx1-C91A (Fig 3D). However, trans-formation of a plasmid encoding a wild-type Prx1 (Prx1-WT) into Δ$prx1$ cells fully restored the $E_{GSH}$ response (Fig 3D). In summary, the most likely explanation for our data is that glutathione is involved in the reduction of Prx1 following its reaction with H$_2$O$_2$, although it is not possible to say whether glutathione is directly or indirectly reducing Prx1 (Greetham & Grant, 2009; Pedrajas *et al*, 2016, 2010; Fig 3E).

## Prx1 hyperoxidation protects matrix glutathione from H$_2$O$_2$-induced oxidation

Given that Prx1 appears to efficiently catalyze the transfer of oxidative equivalents from H$_2$O$_2$ to glutathione, we next asked about the consequences of acute H$_2$O$_2$ challenges for the mitochondrial matrix glutathione pool. To this end, we developed an acute stress-washout assay. In this experiment, Δ$prx1$ cells transformed with a plasmid for the expression of wild-type Prx1 (Δ$prx1$ + Prx1-WT) and

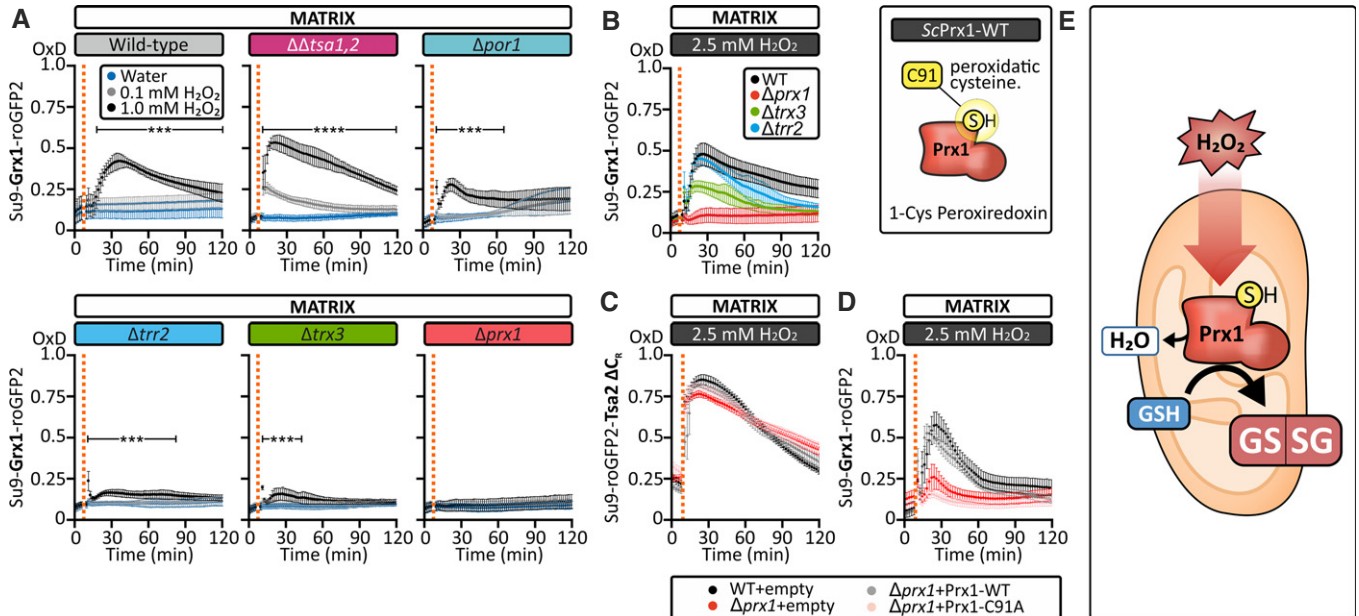

**Figure 3. Prx1 efficiently transfers oxidative equivalents from $H_2O_2$ to glutathione.**

A   The response of a mitochondrial matrix-localized Grx1-roGFP2 sensor expressed in wild-type, $\Delta tsa1\Delta tsa2$, $\Delta por1$, $\Delta trr2$, $\Delta trx3$, or $\Delta prx1$ cells to a bolus of exogenous $H_2O_2$ at the indicated concentrations. Cells were grown in SGal (−Leu) medium and harvested at early exponential phase.

B   Left panel: The response of a mitochondrial matrix-localized Grx1-roGFP2 sensor expressed in wild-type, $\Delta trr2$, $\Delta trx3$, or $\Delta prx1$ cells to a bolus of 2.5 mM $H_2O_2$. Cells were grown in SGal (−Leu) medium and harvested at early exponential phase. Right panel: Scheme depicting Prx1 and its peroxidatic cysteine at position 91.

C   The response of a mitochondrial matrix-localized roGFP2-Tsa2$\Delta C_R$ probe to exogenous 2.5 mM $H_2O_2$. The probe was expressed in wild-type cells transformed with an empty vector or in $\Delta prx1$ cells transformed with either an empty vector, a vector encoding wild-type Prx1, or a vector encoding Prx1-C91A, grown in SGal (−Leu, −Ura).

D   The response to of a mitochondrial matrix-localized Grx1-roGFP2 sensor expressed in either wild-type cells transformed with an empty vector or in $\Delta prx1$ cells transformed with either an empty vector, a Prx1-WT, or a Prx1-C91A plasmid to exogenous $H_2O_2$ at a concentration of 2.5 mM. Cells were grown to SGal (−Leu, −Ura) medium and harvested at early exponential phase.

E   Model illustrating the role of Prx1 in coupling $H_2O_2$ to oxidation of the mitochondrial glutathione pool.

Data information: In all panels, OxD refers to the degree of sensor oxidation. Error bars represent the standard deviation ($n = 3$ biological replicates, with cells obtained from three independent cultures for every strain and probe combinations. For each biological replicate, three technical replicates were performed). Significance was assessed with Student's 2-tailed, unpaired, $t$-test. ***$P < 0.001$ and ****$P < 0.0001$.

expressing matrix Grx1-roGFP2 were incubated with increasing amounts of $H_2O_2$. The $H_2O_2$ was then removed, and in a subsequent readout experiment, the response of the matrix Grx1-roGFP2 probe toward the addition of 1 mM exogenous $H_2O_2$ was monitored (Fig 4A and B). Counterintuitively, we observed a strong negative correlation between the concentration of $H_2O_2$ used in the pre-treatment and the response of the matrix Grx1-roGFP2 to the subsequent bolus of 1 mM exogenous $H_2O_2$ (Fig 4B; for data with wild-type cells, see Appendix Fig S4A). When we repeated the experiment with $\Delta prx1$ cells transformed with a plasmid encoding an enzymatically inactive Prx1-C91A mutant, we observed no response to the 1 mM bolus of exogenous $H_2O_2$, irrespective of the concentration of $H_2O_2$ used in the initial pre-treatment (Fig 4C). To test whether this attenuation of the $E_{GSH}$ response after preceding $H_2O_2$ treatment could also be caused by matrix-originating $H_2O_2$, we employed a matrix-targeted D-amino acid oxidase (DAO; Matlashov et al, 2014). Upon addition of D-alanine but not L-alanine, this enzyme locally produces $H_2O_2$. When we compared the $E_{GSH}$ response toward 1 mM $H_2O_2$, we found that upon pre-treatment with D-alanine but not L-alanine, the $E_{GSH}$ response was indeed attenuated (Fig 4D and Appendix Fig S4B). Thus, both pre-treatment of wild-type cells with

exogenous $H_2O_2$ and matrix-specific generation of $H_2O_2$ induce a $\Delta prx1$-like mitochondrial matrix $E_{GSH}$ response upon subsequent exogenous $H_2O_2$ treatment.

High levels of $H_2O_2$ can lead to hyperoxidation of peroxidatic cysteine thiol groups to sulfinic or sulfonic acids, which render the peroxiredoxin enzymatically inactive. We thus asked whether our $H_2O_2$ pre-treatments lead to hyperoxidation of the Prx1 peroxidatic cysteine. We therefore employed two different assays to test whether Prx1 in the mitochondrial matrix can be hyperoxidized at the concentrations of $H_2O_2$ used in our acute stress assays. First, we made use of a fusion construct between Prx1 and roGFP2, roGFP2-Prx1, which we targeted to the matrix. RoGFP2-based fusion constructs have recently been shown to be well suited for monitoring peroxiredoxin activity and hyperoxidation *in vivo* (Staudacher et al, 2018). In the roGFP2-Prx1 probe, Prx1 will directly interact with $H_2O_2$ and transfer oxidation onto roGFP2. Interaction of roGFP2 with matrix Grx2/glutathione will reduce roGFP2 again. We observed an increasing roGFP2-Prx1 sensor response with increasing concentrations of exogenous $H_2O_2$ up to 2.5 mM (Fig 4E). However, at 5 mM exogenous $H_2O_2$, we observed no further increase in maximum probe oxidation

and an initial more rapid recovery as compared to 2.5 mM $H_2O_2$ (Fig 4E). This is indicative of hyperoxidation of the Prx1 moiety (Staudacher *et al*, 2018). As a control, a matrix-targeted unfused roGFP2 did not exhibit a lowered response with increasing $H_2O_2$ concentrations and showed a similar response to the roGFP2-Prx1 probe at 5 mM $H_2O_2$, further supporting Prx1 inactivation (Fig 4E). Thus, hyperoxidation-based inactivation of the fused peroxiredoxin results in a more reduced roGFP2 (Morgan *et al*, 2016; Roma *et al*, 2018; Staudacher *et al*, 2018). In line with these results, matrix Grx1-roGFP2 responses in a $\Delta tsa1\Delta tsa2$ strain became attenuated upon application of exogenous $H_2O_2$ at

lower concentrations than those used in assays with the wild-type strain (Appendix Fig S4C).

As a second approach to test for hyperoxidation, we used an SDS–PAGE-based redox shift assay to monitor the redox state of the single cysteine in mature endogenous Prx1, i.e., the peroxidatic cysteine C91 (Fig 4F). We used the maleimide compound mmPEG$_{24}$, which upon modification of a cysteine thiol results in an increased mass of the modified protein that can be detected on SDS–PAGE (Kojer *et al*, 2015). Maleimides react with reduced cysteine thiols but not hyperoxidized cysteine residues or cysteine residues involved in disulfide bonds. Samples were either (i) treated with

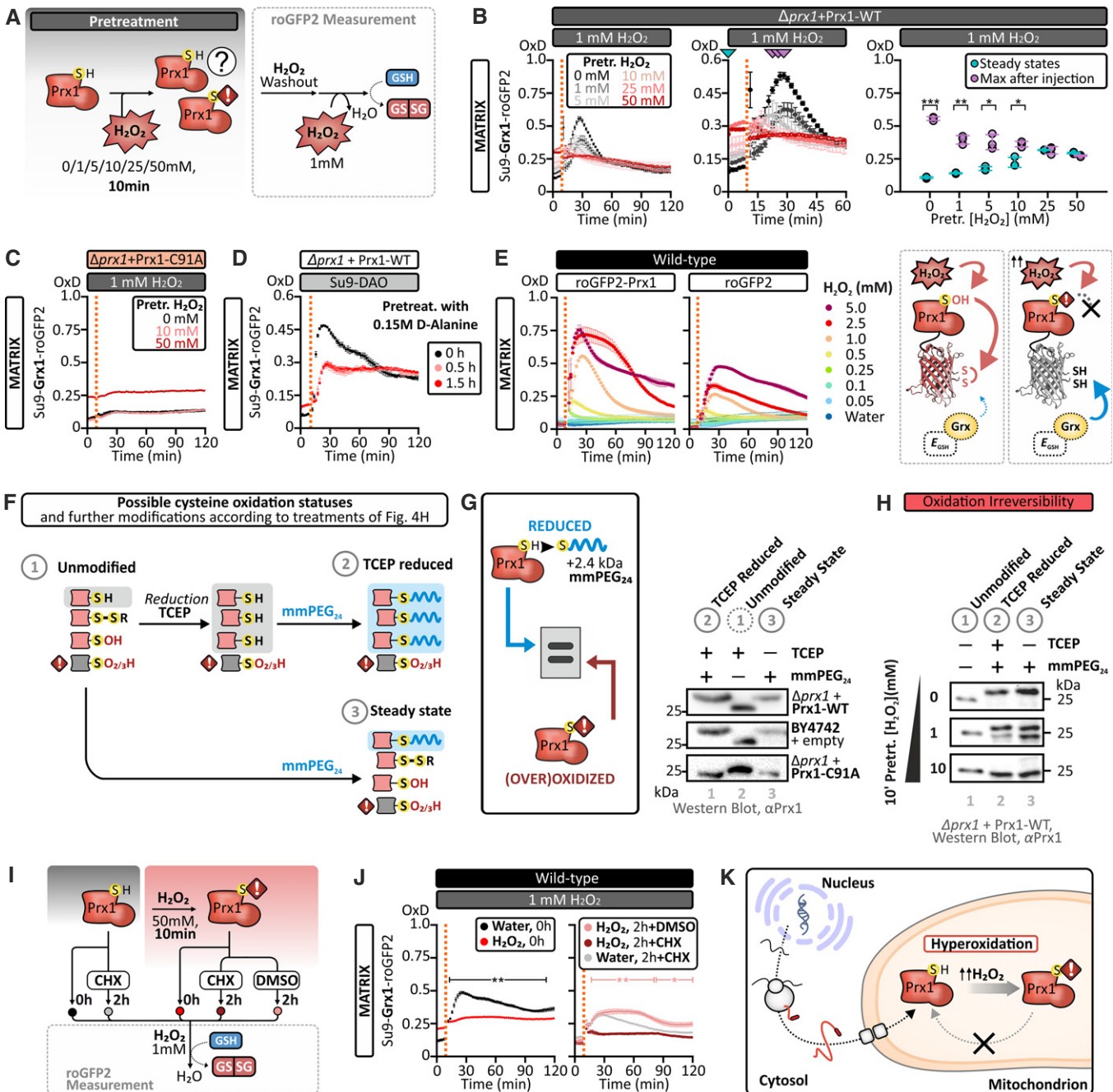

**Figure 4.**

◀

**Figure 4. The active-site cysteine of Prx1 can become hyperoxidized.**

A  Experimental layout of the "acute stress-washout" assay performed in (B) and in Appendix Fig S4A. Cells were pre-treated with either 0, 1, 5, 10, 25, or 50 mM $H_2O_2$ for 10 mins after which $H_2O_2$ was removed, and cells were resuspended in fresh buffer. The response of a mitochondrial matrix-localized Grx1-roGFP2 probe to an exogenous bolus of 1 mM $H_2O_2$ was then measured.

B  "Acute stress-washout" assay. Δprx1 cells containing a Prx1-WT plasmid and expressing a mitochondrial matrix-localized Grx1-roGFP2 probe were grown to exponential phase in SGal (−Leu, −Ura) medium harvested and treated with either 0, 1, 5, 10, 25, or 50 mM $H_2O_2$ for 10 min. After removal of $H_2O_2$ and resuspension in fresh buffer, the response of a mitochondrial matrix-localized Grx1-roGFP2 probe to the addition of 1 mM $H_2O_2$ was measured. The middle panel is an enlargement of the left panel. The graph in the right panel shows the steady-state Grx1-roGFP2 oxidation following $H_2O_2$ pre-treatment as well as the maximum Grx1-roGFP2 oxidation in response to the subsequent second $H_2O_2$ treatment.

C  The same experiment as in (B) performed with Δprx1 cells transformed with a Prx1-C91A plasmid.

D  Δprx1 cells transformed with a vector encoding wild-type Prx1, transformed with a vector encoding the matrix-targeted D-amino acid oxidase (su9-DAO), and expressing a mitochondrial matrix-localized Grx1-roGFP2 were grown to exponential phase in SGal (−Leu, −Ura, −His) and then pre-treated with 0.15 M D-alanine for 0, 0.5, and 1.5 h. Afterward, the cells were washed and the response of Grx1-roGFP2 to the addition of 1 mM $H_2O_2$ was measured. A redox shift assay to establish the redox state of Prx1 in these samples is present in Appendix Fig S4E.

E  *Left panel*: the response of mitochondrial matrix-localized roGFP2-Prx1 and roGFP2 probes expressed in wild-type cells to exogenous $H_2O_2$ at the indicated concentrations. Cells were grown in SGal (−Leu) medium and harvested at early exponential phase. *Right panel*: model illustrating that hyperoxidation of the Prx1 moiety in the roGFP2-Prx1 sensor leads to a roGFP2-like behavior.

F  Scheme illustrating the workflow of the experiment preformed in (G, H). Dependent on the initial state of the single cysteine of Prx1, different outcomes of the modification approach and consequently different behavior on SDS–PAGE can be expected. In sample (1) "unmodified", no shift expected, since no mmPEG$_{24}$ modification occurs. In sample (2) "TCEP-reduced", treatment with TCEP reduces only disulfide bonds and sulfenic acids to thiols and these are modified with mmPEG$_{24}$—hyperoxidized cysteine residues are not modifiable. In lane 3, "steady state", only reduced thiols are directly modified with mmPEG$_{24}$—disulfide bonds, sulfenic acids, and hyperoxidized cysteine residues are not modifiable.

G  *Left panel*: a scheme illustrating the principle of the thiol modification-based "redox shift" assay, for which reduced thiols (but not thiols oxidized to a disulfide bond or a sulfinic or sulfonic acid) can be irreversibly modified by the alkylating agent mmPEG$_{24}$ leading to a ~ 2.4 kDa increase in mass and thus decreased mobility on an SDS–PAGE gel. *Right panel*: redox shift assay to assess the oxidation state of the Prx1 catalytic cysteine residue performed in Δprx1 + Prx1-WT, wild-type + empty vector, and Δprx1 + Prx1-C91A.

H  Redox shift assays of Prx1. Δprx1 cells transformed with a plasmid encoding wild-type Prx1 were grown to early exponential phase in SGal (−Ura) medium. Cells were subsequently treated with the indicated concentrations of $H_2O_2$ for 10 min. Exposure to $H_2O_2$ oxidizes the single cysteine of Prx1 to a state that cannot be reduced with TCEP.

I  Experimental layout of the "acute stress-washout" assay in the presence or absence of the ribosome inhibitor cycloheximide (CHX) as performed in (J). Wild-type cells expressing a mitochondrial matrix-localized Grx1-roGFP2 probe were grown to early exponential phase in SGal (−Leu). Cells were pre-treated either with water or 50 mM $H_2O_2$ for 10 min and afterward washed and resuspended in fresh medium. During the next 2-h recovery, cells were either treated with the translation inhibitor cycloheximide (CHX) or with DMSO as a vehicle control. The response of a mitochondrial matrix-localized Grx1-roGFP2 probe to the subsequent addition of 1 mM $H_2O_2$ was then measured.

J  The response of a mitochondrial matrix-localized Grx1-roGFP2, expressed in wild-type cells grown, to the addition of exogenous 1 mM $H_2O_2$. Cells were grown in SGal (−Leu) and harvested at early exponential phase. *Left panel*: the pre-treatment efficiently inhibits the ability of Prx1 to transfer oxidizing equivalents to $E_{GSH}$. *Right panel*: newly synthesized Prx1 is required to recover the transfer of oxidizing equivalents to $E_{GSH}$ in the mitochondrial matrix.

K  Hyperoxidation of Prx1 can prevent $H_2O_2$-induced oxidation of the matrix glutathione pool, and *de novo* synthesis is required to replace hyperoxidized Prx1.

Data information: In panels (B–E) and (J), OxD refers to the degree of sensor oxidation. Error bars represent the standard deviation ($n = 3$ biological replicates, with cells obtained from three independent cultures for every strain and probe combinations. For each biological replicate, three technical replicates were performed). Significance was assessed with Student's 2-tailed, unpaired, *t*-test. *$P < 0.05$; **$P < 0.01$; and ***$P < 0.001$.

Source data are available online for this figure.

mmPEG$_{24}$ (steady state), (ii) treated with the reducing agent Tris(2-carboxyethyl)phosphine (TCEP) (unmodified), or (iii) treated with both TCEP and mmPEG$_{24}$ (TCEP-reduced). TCEP reduces disulfide bonds and sulfenic acids but not sulfinic or sulfonic acids (Reisz *et al*, 2013). Under unperturbed conditions, the cysteine of Prx1 was found to be predominantly in the thiol/thiolate form and can therefore be modified by maleimide (Fig 4G, lane 3, compare Prx1-WT to Prx1-C91A). Next, we assessed the Cys91 redox state following treatment of cells with $H_2O_2$. We observed that upon treatment with 1 mM $H_2O_2$, half of the Prx1 pool became unreactive toward mmPEG$_{24}$ (Fig 4H, lane 3). This can only be partially reverted by addition of TCEP (Fig 4H, lane 2) indicating partial hyperoxidation of Prx1 already at this $H_2O_2$ concentration. Upon treatment with 10 mM $H_2O_2$, Prx1 is rendered completely unreactive toward mmPEG$_{24}$ (Fig 4H, lane 3), even after subsequent treatment with TCEP, indicative of hyperoxidation of the peroxidatic cysteine. Treatment of matrix DAO-expressing cells with D-alanine but not L-alanine also rendered Prx1 partially unreactive toward mmPEG$_{24}$ (Appendix Fig S4D and E). Thus, exposure of Prx1 to high concentrations of $H_2O_2$ renders its active-site cysteine maleimide

inaccessible (likely hyperoxidized) and results in a lower capacity to elicit an $H_2O_2$-dependent $E_{GSH}$ response.

In the cytosol, hyperoxidation in the form of a sulfinic acid (but not a sulfonic acid) can be reverted by sulfiredoxin. However, sulfiredoxin is not thought to be present in the matrix of yeast mitochondria. We thus tested whether recovery of Prx1 activity after oxidative shock relies on *de novo* Prx1 translation rather than reduction of Cys91. To this end, we allowed cells to recover after acute $H_2O_2$ challenge in the presence or absence of the translation inhibitor cycloheximide (Fig 4I and J). Under these conditions, we observed no recovery of the $E_{GSH}$ response after acute $H_2O_2$ shock in the absence of cytosolic translation, while partial recovery of the $E_{GSH}$ response was observed within 2 h in the absence of cycloheximide (Fig 4J and Appendix Fig S4F). In summary, these results strongly support the hypothesis that hyperoxidation of Prx1 prevents $H_2O_2$-induced oxidation of the matrix glutathione pool. The logical next step was therefore to ask whether Prx1 hyperoxidation and the consequent protection of the mitochondrial glutathione pool have any influence on cell viability under $H_2O_2$ stress.

## Prx1 increases cell sensitivity to acute H$_2$O$_2$ stress

We assessed the importance of Prx1 for cell viability under both chronic and acute H$_2$O$_2$ stresses. Prx1 has previously been shown to be important for maintaining cell viability under chronic oxidative stress (Greetham & Grant, 2009). We confirmed this result using halo assays (Appendix Fig S4G–I). We observed a significant detrimental effect of *PRX1* deletion for growth in the continuous presence of H$_2$O$_2$ (Fig 4I, left panel), while we observed no impact of *PRX1* deletion for viability of cells growing in the continuous presence of the thiol oxidant N,N,N′,N′-tetramethylazodicarboxamide (diamide) (Fig 4I, right panel). Expression of the *Zea mays* aquaporin *Zm*Pip2.5 wild-type but not an inactive mutant, *Zm*Pip2.5 H199K, leads to an increased zone of growth inhibition in our halo assays (Appendix Fig S4H). These results support the idea that aquaporin can mediate the transport of H$_2$O$_2$ across membranes and suggest that in laboratory yeast strains, which typically harbor inactive aquaporins (Laize *et al*, 2000; Sabir *et al*, 2017), H$_2$O$_2$ influx across the plasma membrane is limited. This may partly explain why high concentrations of H$_2$O$_2$ are frequently required to observe effects in yeast assays. We next asked whether *PRX1* deletion also rendered cells more sensitive to an acute H$_2$O$_2$ stress. To this end, we determined the percentage of viable cells following 30-min incubation with H$_2$O$_2$ at concentrations ranging from 0 to 25 mM (Appendix Fig S4J). As a control, we show that expression of the aquaporin *Zm*Pip2.5 wild-type but not the inactive mutant, *Zm*Pip2.5 H199K, led to decreased viability upon acute H$_2$O$_2$ stress (Appendix Fig S4K). Surprisingly, when we compared wild-type and Δ*prx1* cells in these assays, we observed that Δ*prx1* cells were not more sensitive than wild-type cells at H$_2$O$_2$ concentrations below 10 mM, while at higher H$_2$O$_2$ concentrations, the presence of Prx1 was even found to be significantly detrimental (Appendix Fig S4L and M). Thus, Prx1 is not required for cell viability under acute H$_2$O$_2$ stress, and at higher H$_2$O$_2$ concentrations, its presence is detrimental.

The loss of Prx1 causes an adaptive response in the cytosol characterized by increased levels of Ctt1. To test whether these increased Ctt1 levels might explain the superior survival of Δ*prx1* compared to wild-type cells during acute stress, we analyzed a Δ*prx1*Δ*ctt1* strain in the presence or absence of Prx1 complementation (Appendix Fig S4N). Here, we also observed that cells lacking Prx1 performed better than cells containing Prx1 despite the absence of Ctt1 in both strains. Thus, the absence of Prx1 is specifically protective during acute H$_2$O$_2$ stress. Nonetheless, given that Prx1 hyperoxidation at high H$_2$O$_2$ concentration in wild-type cells should effectively mimic a Δ*prx1*-like state, it could be reasonably argued that at high H$_2$O$_2$ concentrations, the acute stress experiment is, in effect, comparing Δ*prx1*-like cells with Δ*prx1* cells. Thus, with these experiments it is not possible to determine the full extent of the protective effect of Prx1 hyperoxidation against acute H$_2$O$_2$ stress-induced cell death.

## Prx1 hyperoxidation protects against H$_2$O$_2$-induced cell death

A rigorous assessment of the role of Prx1 hyperoxidation in acute H$_2$O$_2$ stress-induced cell death requires the development of matrix-targeted Prx1 variants, which are more resistant to hyperoxidation but nonetheless retain their capacity to oxidize glutathione. We were able to generate just such a Prx1 variant, a truncation mutant

of Prx1, Prx1-P233stop (Fig 5A and Appendix Fig S5A). Indeed, we observed an H$_2$O$_2$ concentration-dependent increase in the response of a roGFP2-Prx1-P233stop probe, up to 5 mM exogenous H$_2$O$_2$, in contrast to the decreased response of a roGFP2-Prx1 probe above 2.5 mM exogenous H$_2$O$_2$ (compare Figs 4E and 5B). We further tested this increased resistance of Prx1-P233stop using our gel-based redox shift assay to probe for hyperoxidation. This confirmed that Prx1-P233stop was resistant to hyperoxidation after addition of up to 10 mM exogenous H$_2$O$_2$ (Fig 5C). In the same experiment, we observed hyperoxidation of wild-type Prx1 from 1 mM exogenous H$_2$O$_2$ (Fig 5C). Importantly, by monitoring the response of a matrix-localized Grx1-roGFP2 probe, we observed that the Prx1-P233stop variant was capable of transferring oxidation from H$_2$O$_2$ to glutathione to a similar extent as wild-type Prx1 (Fig 5D, black lines). Furthermore, when we repeated this experiment after an initial pre-treatment of our cells with 10 mM H$_2$O$_2$ for 10 min, we observed that the Grx1-roGFP2 response in Prx1-P233stop-expressing cells was larger than without H$_2$O$_2$ pre-treatment, opposite to what is observed in cells expressing wild-type Prx1 (Fig 5D, red lines). These results are consistent with the strongly decreased hyperoxidation sensitivity of the Prx1P233stop variant, which means that it remains active following the H$_2$O$_2$ pre-treatment. The increased Grx1-roGFP2 response after H$_2$O$_2$ pre-treatment is possibly due to hyperoxidation of cytosolic peroxiredoxins, meaning that more H$_2$O$_2$ reaches the mitochondrial matrix. Additionally, depletion of the matrix NADPH pool in the presence of a non-hyperoxidizable peroxiredoxin might contribute to a stronger Grx1-roGFP2 response. Importantly, we again observed that cells expressing functional Prx1 constructs were significantly more sensitive to acute H$_2$O$_2$ stress-induced cell death than cells lacking Prx1 or expressing an inactive Prx1 variant. The Prx1-P233stop variant also appeared to be more sensitive than Prx1-WT although this difference was statistically significant only at 10 mM H$_2$O$_2$ (Fig 5E and Appendix Fig S5C).

## Mitochondrial glutathione oxidation correlates with cell death under acute H$_2$O$_2$ stress

To further investigate the relationship between matrix glutathione oxidation and cell death, we turned to the peroxiredoxin, *Pf*AOP, from the parasite *Plasmodium falciparum*. *Pf*AOP is known to efficiently transfer oxidation to glutathione and has a well-characterized mutant, L109M (Staudacher *et al*, 2015, 2018; Appendix Fig S6A). *In vitro*, L109M has an increased activity and decreased susceptibility to hyperoxidation compared to wild-type *Pf*AOP. Under the conditions of our assay, *Pf*AOP-L109M indeed more efficiently oxidized $E_{GSH}$ than wild-type *Pf*AOP. However, like the wild-type cells it became inactivated by an acute H$_2$O$_2$ pre-incubation (Appendix Fig S6B). Consistent with our previous results, we observed that cells expressing *Pf*AOP-L109M were significantly more sensitive to acute H$_2$O$_2$ treatment-induced cell death than cells expressing Prx1 and much more sensitive than cells with no matrix-localized thiol peroxidase (Fig 6A and Appendix Fig S6C). *Pf*AOP-L109M-expressing cells also appeared to be more sensitive to acute H$_2$O$_2$ treatment-induced cell death than wild-type *Pf*AOP-expressing cells although these differences were not statistically significant.

Finally, we turned to targeting Tsa1, human PRDX3, and human PRDX5 (which are the two 2-Cys peroxiredoxins residing in the matrix of human mitochondria) as well as human PRDX6 (a 1-Cys peroxiredoxin found in the cytosol of human cells) to the

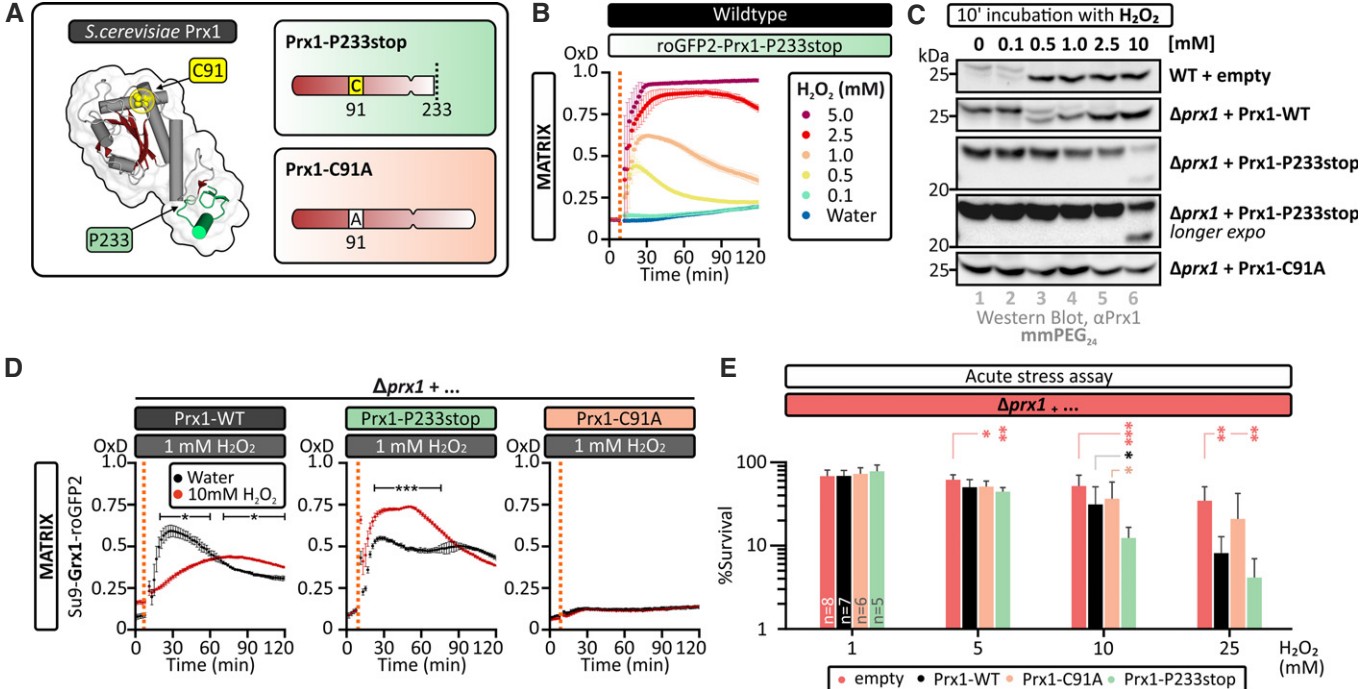

**Figure 5. Hyperoxidation of Prx1 promotes cell survival upon acute H₂O₂ stress.**

A  Structure of *Sc*Prx1 (PDB ID: 5YKJ). The positions of relevant amino acids are indicated, and the applied modifications in Prx1-P233stop and Prx1-C91A are shown. The C-terminal region absent in the Prx1-P233stop mutant is highlighted in green.

B  The response of a mitochondrial matrix-localized roGFP2-Prx1-P233stop probe expressed in wild-type cells to the indicated concentrations of exogenous H₂O₂. Cells were grown in SGal (−Leu) medium and harvested at early exponential phase. Error bars represent the standard deviation (*n* = 3 biological replicates, with cells obtained from three independent cultures for every strain and probe combinations. For each biological replicate, three technical replicates were performed).

C  Maleimide-based gel shift assay to assess Prx1 cysteine redox state. Wild-type cells transformed with an empty plasmid or Δ*prx1* cells with either an empty plasmid or a plasmid encoding wild-type Prx1, the hyperoxidation-resistant Prx1-P233stop, or the inactive Prx1-C91A variants were grown to exponential phase in SGal (−Ura) medium. Cells were subsequently treated with the indicated concentrations of H₂O₂ for 10 min and then directly treated with the alkylating agent mmPEG₂₄.

D  The response of a mitochondrial matrix-localized Grx1-roGFP2 probe, expressed in Δ*prx1* cells containing a plasmid encoding either wild-type Prx1, Prx1-P233stop, or Prx1C91A, to the addition of 1 mM exogenous H₂O₂. Cells had been pre-treated with either 10 mM H₂O₂ or water as a control. For these experiments, cells were grown in SGal (−Leu, −Ura) medium and harvested at early exponential phase. OxD refers to the degree of sensor oxidation. Error bars represent the standard deviation (*n* = 3 biological replicates, with cells obtained from three independent cultures for every strain and probe combinations. For each biological replicate, three technical replicates were performed).

E  Hydrogen peroxide "acute stress" assay. Δ*prx1* cells co-transformed with empty vector or with a vector encoding either wild-type Prx1, the P233stop, or the C91A variants were grown in SD (−Ura) and treated with the indicated concentrations of H₂O₂ for 30 min. Afterward, the cells were diluted, and a fixed volume was plated on YPD plates. The number of viable colonies was counted after 2 days growth at 30°C, here represented as a percentage relative to the 0 mM treatment. Error bars represent standard deviation (*n* = 5–8 biological replicates, with cells taken from independent cultures for each individual biological replicate).

Data information: In all relevant panels, significance was assessed with Student's 2-tailed, unpaired, *t*-test. *P < 0.05; **P < 0.01; and ***P < 0.001.
Source data are available online for this figure.

mitochondrial matrix of Δ*prx1* cells (Appendix Fig S6A). We found that Tsa1 only very inefficiently transferred oxidation to glutathione, consistent with thioredoxin being the preferred reductive partner for this protein (Tairum *et al*, 2012). A similar result was observed for PRDX3 and PRDX5 (Appendix Fig S6B). In all three cases, a slightly increased Grx1-roGFP2 response was observed in cells pre-treated with 10 mM H₂O₂, implying that at least a fraction of these proteins remains in a non-hyperoxidized state (Appendix Fig S6B). In contrast, PRDX6 led to efficient oxidation of glutathione based on the matrix Grx1-roGFP2 response. Furthermore, H₂O₂ pre-treatment further increased this response, again likely due to hyperoxidation-based inactivation of cytosolic peroxiredoxins (Appendix Fig S6B). Strikingly, cells expressing PRDX6 were significantly more sensitive to acute H₂O₂ stress-induced cell death compared to cells expressing

Prx1 and particularly compared to Δ*prx1* cells transformed with PRDX5, Tsa1, or an empty plasmid (Fig 6A and Appendix Fig S6C). Overall, while cells expressing mitochondria-localized Tsa1, PRDX3, PRDX5, or only containing an empty plasmid remained up to 60% viable after 30-min treatment with 10 mM H₂O₂, cells expressing Prx1 were ~ 25% viable, while those expressing *Pf*AOP-L109M or PRDX6 were only ~ 10% viable.

$E_{GSH}$ responses and acute stress assays indicated that PRDX6 is hyperoxidation-resistant. To directly test this, we performed the maleimide shift assay with PRDX6 in comparison with Prx1 (Appendix Fig S6D–F). We observed that PRDX6 cysteines became inaccessible to mmPEG modification at comparatively low H₂O₂ concentrations. However, by treating with TCEP before maleimide modification, we found that a large fraction of PRDX6 oxidation was

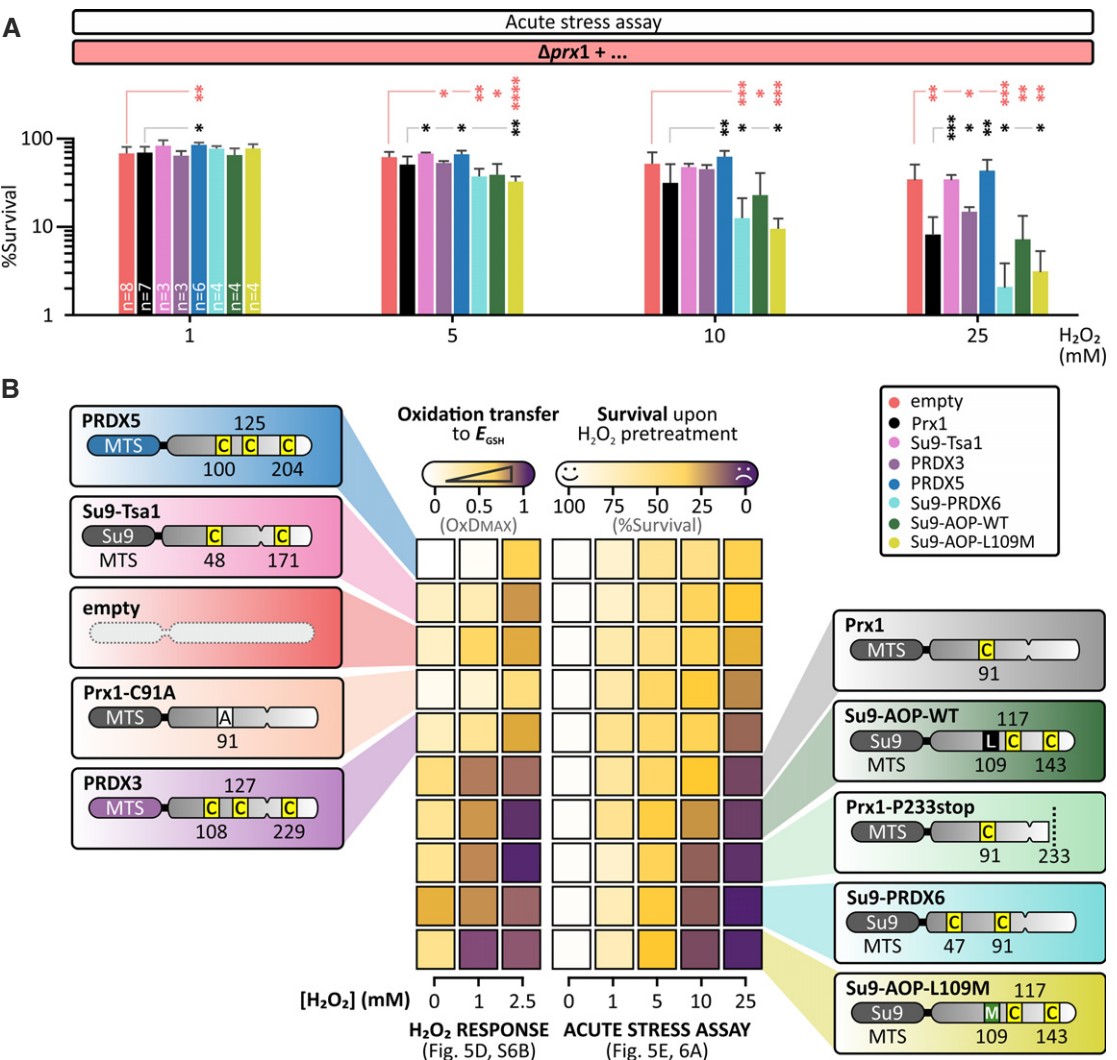

**Figure 6. Modulating the coupling between H₂O₂ and matrix glutathione links matrix glutathione oxidation to cell death.**

A Hydrogen peroxide "acute stress" assay. Δ*prx1* cells co-transformed with an empty plasmid or a plasmid encoding Prx1, Tsa1, PRDX3, PRDX5, PRDX6, *Pf*AOP, or *Pf*AOP-L109M were grown in SD medium lacking the appropriate amino acids for plasmid selection and pre-treated with 0, 1, 5, 10, and 25 mM H₂O₂ for 30 min. Afterward, the cells were diluted, and a fixed volume was plated on YPD plates. The number of viable colonies was counted after 2 days of growth at 30°C, here represented as a percentage relative to the 0 mM pre-treatment. Error bars represent standard deviation (*n* = 3–8 biological replicates, with cells taken from independent cultures for each individual biological replicate). Significance was assessed with Student's 2-tailed, unpaired, *t*-test. \**P* < 0.05; \*\**P* < 0.01; and \*\*\**P* < 0.001.

B The efficiency of transfer of oxidation from H₂O₂ to $E_{GSH}$ strongly correlates with cell death upon exposure to acute H₂O₂ stress. Data from viability and matrix Grx1-roGFP2 responses during acute H₂O₂ stress were correlated.

reversible and could therefore conclude that PRDX6 was not hyperoxidized, indicating that PRDX6 is strongly resistant against hyperoxidation. In conclusion, a lack of hyperoxidation sensitivity, which means that a peroxiredoxin remains active under acute H₂O₂ stress, leads to increased cell death probably due to oxidation of matrix glutathione (Fig 6B).

**Mitochondrial glutathione oxidation is the predominant determinant of cell death following acute H₂O₂ stress**

Is glutathione oxidation the main determinant for cell death or just a proxy for depletion of matrix NADPH levels? To answer this question, we first assessed the impact of *GLR1* deletion on matrix $E_{GSH}$, with and without concomitant deletion of *PRX1*. In Δ*glr1* cells, GSSG reduction is impaired, thereby preserving the NADPH pool (at the expense of a more oxidized $E_{GSH}$), while in Δ*prx1* cells, GSH oxidation is impaired (preserving both the NADPH pool and $E_{GSH}$). In line with this, we observed a strongly increased matrix $E_{GSH}$ response to exogenous H₂O₂ in Δ*glr1* cells compared to wild-type cells, which could not be rescued by expression of only the cytosolic form of Glr1. However, upon additional deletion of *PRX1*, to generate Δ*glr1*Δ*prx1* cells expressing only cytosolic Glr1, this increased matrix Grx1-roGFP2 response was almost completely absent (Fig 7A). We next assessed the growth of these cells. Intriguingly,

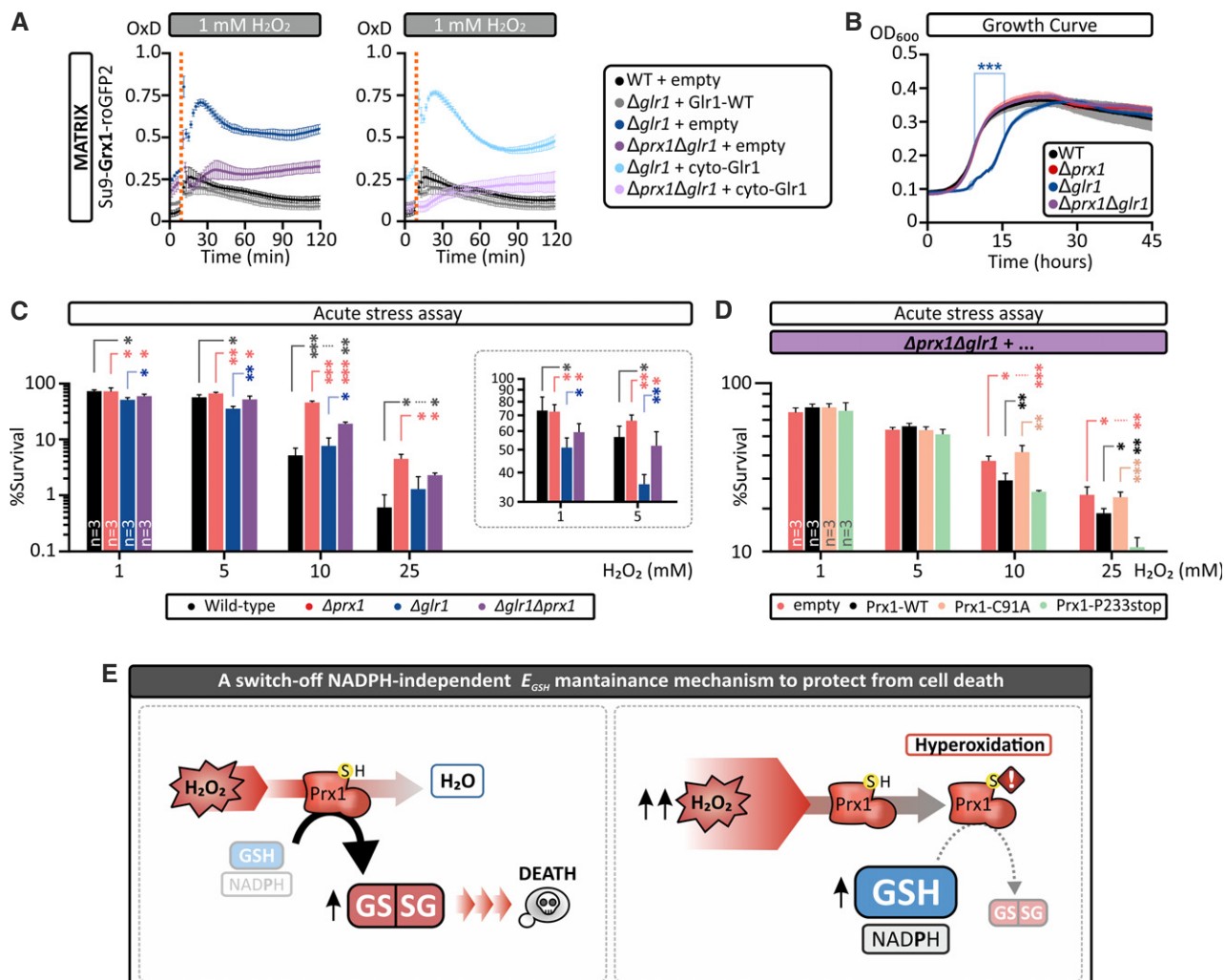

**Figure 7. Mitochondrial matrix GSSG accumulation drives cell death during acute $H_2O_2$ stress.**

A The response of a mitochondrial matrix-localized Grx1-roGFP2 probe to 1 mM $H_2O_2$ in BY4742 wild-type cells with an empty vector, in $\Delta glr1$ cells transformed either with an empty vector or with a vector encoding wild-type Glr1 or the cytosolic form of Glr1, where the MTS-encoding region was removed, or in $\Delta glr1\Delta prx1$ cells transformed either with an empty vector or with a vector encoding the cytosolic form of Glr1. Cells were grown to exponential phase in SGal (−Leu, −Ura) medium. Error bars represent the standard deviation ($n$ = 3 biological replicates, with cells obtained from three independent cultures for every strain and probe combinations. For each biological replicate, three technical replicates were performed).

B Growth curve of wild-type, $\Delta prx1$, $\Delta glr1$, and $\Delta glr1\Delta prx1$ cells in SD medium complemented with all amino acids ($n$ = 3 biological replicates). Significance for the difference in the time the cultures reach 50% of their maximal $OD_{600}$ was assessed with the $t$-test. Error bars (as ribbon) represent the standard deviation ($n$ = 4 biological replicates, with cells taken from independent biological replicate).

C $H_2O_2$ "acute stress" assay. Wild-type, $\Delta prx1$, $\Delta glr1$, and $\Delta glr1\Delta prx1$ cells pre-grown in SGal medium complemented with all amino acids to early exponential phase. Cells were treated with $H_2O_2$ at the indicated concentrations for 30 min. Subsequently, cells were diluted, and a fixed volume was plated on YPD plates. The number of viable colonies was counted after 2 days growth at 30°C, here represented as a percentage relative to the 0 mM pre-treatment. Error bars represent standard deviation ($n$ = 3 biological replicates, with cells taken from independent cultures for each individual biological replicate). An inset with a different scale on the $y$-axis is presented for the 1 and 5 mM concentration to allow better interpretation.

D $H_2O_2$ "acute stress" assay. $\Delta prx1\Delta glr1$ cells co-transformed with an empty plasmid or with a plasmid encoding either wild-type Prx1, the P233stop or the C91A variants were grown in SGal (−Ura) medium to early exponential phase. Cells were treated with the indicated amounts of $H_2O_2$ for 30 min. Subsequently, the cells were diluted, and a fixed volume was plated on YPD plates. The number of viable colonies was counted after 2 days growth at 30°C, here represented as a percentage relative to the 0 mM treatment. Error bars represent standard deviation ($n$ = 3 biological replicates, with cells taken from independent cultures for each individual biological replicate).

E Model. Prx1 drives oxidation of glutathione upon exposure to hydrogen peroxide. During acute $H_2O_2$ stress, Prx1 becomes hyperoxidized, which effectively uncouples the glutathione pool from $H_2O_2$. Prx1 hyperoxidation thus helps to limit GSSG accumulation and contributes to cell survival.

Data information: Significance was assessed with Student's 2-tailed, unpaired, $t$-test. *$P < 0.05$; **$P < 0.01$; and ***$P < 0.001$.

we observed that $\Delta glr1$ cells had an extended lag phase, while the additional deletion of *PRX1* ($\Delta glr1\Delta prx1$ cells) rescued this growth delay (Fig 7B). Lastly, we tested these cells in the acute $H_2O_2$ stress

assay. We found that $\Delta glr1$ cells were severely impaired in their survival after acute stress (Fig 7C, 1 and 5 mM $H_2O_2$). However, the additional deletion of *PRX1* improved survival during acute $H_2O_2$

stress. Complementation of this double deletion strain with different Prx1-WT, Prx1-C91A, and Prx1-P233stop variants confirmed that an increased capacity to promote glutathione oxidation during acute $H_2O_2$ stress resulted in decreased cell survival (Fig 7D). Collectively, these data thus support the conclusion that matrix glutathione oxidation specifically promotes cell death (Fig 7E).

## Discussion

### Prx1 activity leads to the oxidation of glutathione

Prx1 is a 1-Cys peroxiredoxin in the Prx6 subfamily (Nelson *et al*, 2011). The physiological reductive mechanism of such peroxiredoxins remains unclear. Studies on Prx1 from *S. cerevisiae* have differentially concluded that: (i) GSH is part of the reductive mechanism of Prx1 but is not oxidized in the process, i.e., GSH serves as a "resolving cysteine", forming a disulfide bond with the Prx1 peroxidatic cysteine, which is reduced by Trx3 (Pedrajas *et al*, 2016); (ii) GSH forms a transient mixed disulfide bond with the Prx1 peroxidatic cysteine that is subsequently reduced by Grx2, ultimately leading to GSSG formation (Pedrajas *et al*, 2010); (iii) GSH forms a transient mixed disulfide with the Prx1 peroxidatic cysteine that is subsequently reduced by Trr2, leading to the formation of a transient intermolecular disulfide between Prx1 and Trr2 that is reduced by GSH leading to GSSG formation (Greetham & Grant, 2009); and (iv) *in vitro*, the mitochondrial thioredoxin system, Trx3 and Trr2, can efficiently reduce Prx1 (Pedrajas *et al*, 2000).

While the study of the Prx1 reductive mechanism was not a primary objective of our study, our data unequivocally show that, inside living cells, Prx1 activity very efficiently drives GSSG formation. While it is not possible to conclude whether the involvement of GSH in Prx1 reduction is direct or indirect, our data nonetheless allow us to exclude models that suggest no involvement of GSH in Prx1 reduction. The involvement of GSH in the reduction of Prx6-type peroxiredoxins appears not to be restricted to yeast Prx1 as human PRXD6, targeted to the yeast mitochondrial matrix, also very efficiently elicited GSSG formation upon treatment of cells with $H_2O_2$. This observation is interesting as previously PRDX6 was suggested to require a GST-Pi to mediate its enzymatic activity (Manevich *et al*, 2004; Zhou *et al*, 2013). Our observations show that GST-Pi cannot be essential for PRDX6 activity as there is no GST-Pi in the yeast matrix, although it cannot be completely excluded that other enzymes in the yeast matrix may fulfill a similar role.

The observation that $H_2O_2$-induced GSH oxidation requires Prx1 also supports the more general assertion that $H_2O_2$-induced oxidation of most cellular thiols requires enzymatic catalysis. Indeed, cytosolic thioredoxin oxidation under acute $H_2O_2$ stress was also shown to be dependent on the presence of cytosolic peroxiredoxins (Day *et al*, 2012). Furthermore, it seems increasingly likely that $H_2O_2$-induced oxidation of many, if not most, $H_2O_2$-sensitive protein thiols requires enzyme catalysis, namely by thiol peroxidases (Delaunay *et al*, 2002; Bozonet *et al*, 2005; Jarvis *et al*, 2012; Sobotta *et al*, 2015; Stocker *et al*, 2018). Together, these studies underline the importance of kinetics and kinetic (un)coupling in determining the relative poise of different cellular redox species and redox couples and underline the conclusion that there is no such thing as a general cellular or subcellular "redox state".

### Mitochondrial glutathione oxidation leads to cell death

In this study, we have demonstrated that oxidation of the mitochondrial glutathione pool, driven almost exclusively by Prx1 activity, is an important determinant of cell death upon exposure to an acute $H_2O_2$ stress (Fig 7E). The groups of Elizabeth Veal and Chris Grant have previously reported that oxidation of cytosolic thioredoxins in fission yeast and oxidation of the mitochondrial thioredoxin in budding yeast, respectively, correlate with $H_2O_2$-induced cell death (Day *et al*, 2012; Greetham *et al*, 2013). These studies are in line with our findings and suggest that depletion of cellular reductive systems is an important determinant of cell death upon an acute $H_2O_2$ stress and not, as might have previously been expected, an accumulation of "oxidative damage". Nonetheless, the conclusions from these different studies raise the question of which, if any, redox couples are directly leading to cell death, i.e., is oxidation of thioredoxins or glutathione important, or are both perhaps simply markers of NADPH depletion. Furthermore, it may also be asked whether oxidation in the cytosol or mitochondrial matrix is more important for triggering cell death.

To address the above questions, we have employed several assays to allow us to distinguish between the relative importance of different matrix redox couples. First, by employing an matrix "redox engineering" approach in which we targeted different peroxiredoxins, with differing sensitivities to hyperoxidation and differing abilities to transfer oxidation to glutathione, we revealed a strong correlation between matrix glutathione oxidation and cell death upon acute $H_2O_2$ stress. The use of engineered peroxiredoxins (even from other species and compartments) in this study also argues against Prx1 exerting a specific intrinsic cell death signaling function. Second, to enable us to distinguish between glutathione oxidation and other possible causes of cell death, for example, NADPH depletion, we turned to strains deleted for glutathione reductase. Specifically, we observed that deletion of the matrix form of glutathione reductase leads to matrix glutathione oxidation, even under "unperturbed" conditions, and leads to a delayed entry to exponential growth. Intriguingly, we found that concomitant deletion of *PRX1* in this background restored matrix $E_{GSH}$ to a wild-type-like value and completely rescued the delayed cell growth. Furthermore, we found that Δ*glr1* cells fared much worse that wild-type cells at 1 and 5 mM exogenous $H_2O_2$ in an acute $H_2O_2$ stress assay. Importantly however, the additional deletion of *PRX1* at least partially rescued the increased sensitivity of a Δ*glr1* strain. As deletion of *GLR1* would not be expected to decrease NADPH levels, these assays support the hypothesis that matrix glutathione oxidation specifically is detrimental for cell fitness. Nevertheless, it is important to emphasize that we cannot completely exclude that NADPH depletion may also be an important determinant of cell death upon acute $H_2O_2$ stress. Perhaps both matrix GSSG accumulation and NADPH depletion synergistically lead to cell death under many circumstances. However, our evidence does clearly support a role for matrix GSSG accumulation in driving cell death that is additional to, and independent of, any role of NADPH depletion. This conclusion is also consistent with a previous study that reported that the increased oxidant sensitivity of yeast cells lacking glutathione reductase can be ascribed to the loss of matrix glutathione reductase activity (Gostimskaya & Grant, 2016).

**Is Prx1 hyperoxidation physiologically relevant?**

In wild-type cells, it was proposed that Prx1 hyperoxidation represents an important protective mechanism to mitigate against oxidation of the matrix glutathione pool in conditions of acute $H_2O_2$ stress. However, this leads to the question of whether peroxiredoxin oxidation is physiologically relevant, i.e., will cells ever encounter conditions severe enough to induce peroxiredoxin oxidation during their "normal" lifestyle?

It is our contention that the almost universal conservation of hyperoxidation sensitivity-associated GGLG and YF motifs among eukaryotic typical 2-Cys peroxiredoxins argues that there is a strong evolutionary selective pressure to evolve or maintain hyperoxidation sensitivity (Wood *et al*, 2003; Hall *et al*, 2009). Indeed, many bacterial typical 2-Cys peroxiredoxins are robust against hyperoxidation, without any apparent detrimental impact upon catalytic efficiency (Ferrer-Sueta *et al*, 2011; Perkins *et al*, 2014). One possible reason for maintaining hyperoxidation sensitivity is that hyperoxidized peroxiredoxins serve specific functions within the cell, for example, recruitment of chaperones to protein aggregates (Hanzen *et al*, 2016) or prevention of excessive oxidation of important cellular redox couples and therefore maintenance of cell viability, as shown in this study and in others (Day *et al*, 2012; Veal *et al*, 2018). Other conserved motifs were recently shown to further fine-tune peroxiredoxin hyperoxidation sensitivity (Bolduc *et al*, 2018), likely serving to tailor each peroxiredoxin to its specific function(s), localization, and the typical $H_2O_2$ concentrations that it will encounter. In the specific case of Prx1, the possibility to generate a hyperoxidation-insensitive variant, without any apparent loss in catalytic prowess, by removal of part of the C-terminus (P233stop) implies that specific motifs also exist in Prx6-family, 1-Cys peroxiredoxins that confer sensitivity to hyperoxidation. However, the identity of such motifs was beyond the scope of this study and remains completely unclear. For Prx1, it remains to be demonstrated to what extent hyperoxidation occurs under "normal" physiological conditions in yeast cells grown in standard laboratory conditions. However, it is important to note that our data show a progressive increase in Prx1 hyperoxidation with increasing $H_2O_2$ concentration. We observed Prx1 hyperoxidation with as little as 1 mM exogenous $H_2O_2$: it is unclear what this means in terms of mitochondrial matrix $H_2O_2$ levels, but likely they are at most in the low micromolar range in these conditions. Thus, Prx1 hyperoxidation is not a "digital" all-or-nothing switch but rather an "analog" switch, implying that hyperoxidation of even a fraction of Prx1 may be beneficial for cell survival and therefore arguing strongly in favor of the physiological relevance of this mechanism. Investigation of the possible biological reasons for the wide range of peroxiredoxin hyperoxidation sensitivities will undoubtedly be an interesting area for future research.

As a final note, caution should always be applied when interpreting results obtained with laboratory yeast strains, grown in a carefully controlled environment. It is unclear what "normal physiological conditions" means for a yeast strain living in its wild environment. The impact of environmental stresses on cellular redox homeostasis, for example, heat, desiccation, UV radiation, limitation of key nutrients and exposure to xenobiotic compounds, is barely understood. Furthermore, certain potentially $H_2O_2$ relevant proteins, for example, aquaporins, are typically inactive in many laboratory yeast strains (Laize *et al*, 2000; Sabir *et al*, 2017). This

adaptation, likely to frequent freezing and thawing, might explain an increased resistance to externally added $H_2O_2$. In line with this, artificial expression of aquaporins in the yeast plasma membrane (Appendix Fig S4) rendered these cells more sensitive to chronic and acute $H_2O_2$ stresses. It is thus tempting to speculate that hyperoxidation may be a highly relevant phenomenon for true wild-type cells in their native environment.

**Compartment-specific NADPH-independent systems to regulate $E_{GSH}$**

$E_{GSH}$ is robustly maintained in most investigated compartments (Meyer *et al*, 2007; Kojer *et al*, 2012; Morgan *et al*, 2013; Elbaz-Alon *et al*, 2014). In the cytosol and the matrix, NADPH-dependent glutathione reductase provides the major mechanism of $E_{GSH}$ maintenance. Impairment of glutathione reductase in the cytosol is compensated by crosstalk between the glutathione and thioredoxin redox systems, and a conserved NADPH-independent $E_{GSH}$ maintenance pathway, i.e., the ABC transporter-dependent removal of GSSG from the cytosol (Minich *et al*, 2006; Morgan *et al*, 2013). In the matrix, GSSG accumulation is prevented under conditions of acute stress by switching off Prx1 activity, which appears to be the exclusive source of GSSG upon $H_2O_2$ exposure. Inactivation of Prx1 is achieved by apparently irreversible hyperoxidation that can only be reversed by synthesizing and importing new Prx1. This indicates that under acute stress conditions, the prime imperative for the matrix seems to be avoiding excessive oxidation of the glutathione pool rather than the immediate removal of $H_2O_2$, thereby ensuring cell viability.

**A cytosolic adaptive response occurs upon matrix redox enzyme perturbation**

Restricting the amounts of $H_2O_2$, which reach the matrix by efficient cytosolic $H_2O_2$ removal, appears to be a further strategy of mediating compartment-specific redox homeostasis. Under unperturbed conditions, the 2-Cys peroxiredoxins Tsa1 and Tsa2 perform this task. By employing transcriptome analyses, we also found evidence for a cytosolic adaptive response, which is induced under conditions of impaired function of matrix redox enzymes. This adaptive response specifically involves upregulation of the catalase, Ctt1, thereby providing an additional NADPH-independent system of $H_2O_2$ handling to further decrease the amount of $H_2O_2$ reaching the matrix. It will be extremely interesting to identify the signal that allows for a nuclear transcriptional response upon disruption of matrix redox enzymes. It is tempting to speculate that there is a specific target protein involved in the signaling mechanism whose activity is regulated by a Prx1-dependent post-translational redox modification.

# Materials and Methods

All reagents were purchased from Sigma-Aldrich unless otherwise stated.

**Generation and growth of yeast strains**

All experiments were performed in a BY4742 strain background (EUROSCARF, Frankfurt, Germany). Strains are listed in

Appendix Table S1. Gene deletion strains were constructed using a PCR-based standard homologous recombination technique (Janke *et al*, 2004). Gene deletions were confirmed by PCR using primers designed to anneal ~ 100 bp up- and downstream of the gene of interest. Yeast strains were grown as described previously (Kojer *et al*, 2012). Briefly, for the roGFP2-based experiments and the redox shift experiments, the strains were grown in synthetic medium (S-medium) lacking the appropriate amino acids for plasmid selection (given in the standard three-letter code, e.g., S-medium—Leu) with 2% galactose (Gal) as carbon source at 30°C. The medium used for growth of strains is given for each experiment in the Figure Legends. As an example, "SGal-(Leu)" indicates synthetic medium with galactose as carbon source and lacking the metabolic marker amino acid leucine. For the acute stress experiments, the strains were grown in synthetic medium lacking the appropriate amino acids for plasmid selection with 2% glucose (D) or 2% galactose (Gal) as carbon source. After $H_2O_2$ exposure, cells were plated on rich media plates with glucose as carbon source (YPD). Growth to assess cell viability following the stress assays took place at 30°C for 2 days.

For 1 l synthetic medium (S-medium), 1.7 g yeast nitrogen base (without amino acids) and 5 g ammonium sulfate were dissolved in water at pH 5.5. In the final medium, as carbon source either 2% glucose (D), 2% galactose (Gal), or 2% glycerol (G) was present. In addition, according to the auxotrophic selections, the following amino acids were either present or excluded: adenine (0.15 mM), lysine (0.20 mM), leucine (0.23 mM), histidine (0.10 mM), tryptophan (0.07 mM), and uracil (0.18 mM). For 1 l rich medium (yeast–peptone, YP-medium), 10 g yeast extract and 20 g bacto-peptone were dissolved in water at pH 5.5. In the final medium, as carbon source either 2% glucose (D), 2% galactose (Gal), or 2% glycerol (G) was present. All plates have been prepared following the recipes described above and adding agar 20 g/l.

**Primers and plasmid construction**

All roGFP2 sensors were constructed as previously described (Gutscher *et al*, 2008; Morgan *et al*, 2016). All roGFP2-fusion proteins used in this study were constitutively expressed from the low copy-number (*CEN*) p415 (*LEU2* marker) or p416 plasmids (*URA3* marker) under the control of a constitutive TEF promoter (from translation elongation factor 1α gene, TEF2, yeast). For mitochondrial matrix targeting, indicated constructs were genetically fused with the N-terminal mitochondrial targeting sequence (MTS) from subunit 9 of the $F_0$-ATPase (Su9) from *Neurospora crassa* (encoding amino acids 1–69).

The mitochondrial peroxiredoxin *PRX1* gene was amplified by PCR including endogenous promoter and terminator from yeast genomic DNA preparations and cloned into the low copy-number (*CEN*) p416 (*URA3* marker) vector using standard molecular cloning procedures, removing the TEF promoter and CYC terminator from the plasmid (pEPT *PRX1(WT)*). The *PRX1(P233stop)* and the *PRX1 (C91A)* constructs were generated from the *PRX1(WT)* construct by site-directed mutagenesis.

The open reading frame of *GLR1* and *TSA1* gene was amplified by PCR from yeast genomic DNA and cloned into the p416 vector with TEF promoter and C-terminal triple hemagglutinin (HA) tag.

To allow mitochondrial targeting, the *TSA1* gene was genetically fused with the N-terminal Su9 MTS.

The AOP constructs from *P. falciparum* were constructed as previously described (Staudacher *et al*, 2018). They were genetically fused with the N-terminal Su9 MTS and expressed in the low copy-number p416 vector (*URA3* marker) with TEF promoter.

The genes of *Homo sapiens* PRDX3, PRDX5, and PRDX6 were amplified by PCR from HEK293 cell cDNA preparations and cloned, respectively, into the low copy-number p416 (*URA3* marker), p415 (*LEU2* marker), and p413 (*HIS3* marker) vectors with TEF promoter and C-terminal triple hemagglutinin (HA), hexahistidine (His6), or FLAG tags. Only the PRDX6 gene was genetically fused with the N-terminal Su9 MTS, while for PRDX3 and PRDX5 genes, their endogenous MTS was used for mitochondrial targeting.

The constructs for the yeast-codon-optimized aquaporin PIP25 wild-type and H199K variants from *Z. mays* were previously described (Bienert *et al*, 2014). They were expressed in the multi-copy pRS426-pTPIu (*URA3* marker) vector with TPI promoter (from triosephosphate isomerase gene, TPI1, yeast).

The gene of D-amino acid oxidase (DAO) from *Rhodotorula gracilis* (red yeast) was amplified by PCR from the "HyPer-D-amino acid oxidase" plasmid generated and characterized in Matlashov *et al* (2014). To allow mitochondrial targeting, the PCR product was genetically fused with the N-terminal Su9 MTS and C-terminal FLAG tag; subsequently, it was cloned into the low copy-number p413 (*HIS3* marker) vector under the control of the strong constitutive GPD promoter (from glyceraldehyde-3-phosphate dehydrogenase gene, TDH3/GPD, yeast). For a detailed list of primers and plasmids, see Appendix Table S2.

**Antibodies**

The antibody against yeast Prx1 was generated in this study. Mature Prx1 without the MTS was subcloned into pRSET-A plasmid. Prx1 was expressed for 4 h at 30°C in *Escherichia coli* BL21 (DE3) cells, purified via its N-terminal hexahistidine (His6) tag and used for in-house immunization of rabbits. The serum was confirmed in immunoblot analysis using the purified antigen as well as comparing yeast cells lacking Prx1 with wild-type cells. The serum was used in a 1:1,000 dilution in 5% Milk-TBS for Western blot detection (Milk powder 5%, NaCl 150 mM, Tris–HCl 10 mM, adjusted to pH 7.5).

The antibody against PRDX6 was purchased from Sigma (ID: P0058) and used 1:2,000 dilution in 5% milk–TBS for Western blot detection.

**Fluorescence measurement of roGFP2-sensor oxidation**

Fluorescence measurements were performed with a CLARIOstar (BMG Labtech) fluorescence plate reader as described previously (Morgan *et al*, 2011). Fluorescence was recorded using filter optics at excitation wavelengths 410 ± 5 and 482 ± 8 nm and an emission wavelength of 530 ± 20 nm. For measurements, yeast cells were grown to mid-log phase in synthetic medium lacking the appropriate amino acids for plasmid selection. The cells were harvested by centrifugation at 1,500 × *g* for 3 min at room temperature and subsequently resuspended in an isosmotic buffer (sorbitol 0.1 M, Tris–HCl pH 7.4 0.1 M, NaCl 0.1 M) to a final concentration of 1.5 $OD_{600}$ units/ml (where 1 $OD_{600}$ unit represents 1 ml of culture with

an $OD_{600} = 1$). This process was repeated one more time. Subsequently, each cell solution was transferred to a flat-bottomed 96-well imaging plate (BD Falcon) with 180 μl solution per well. To one well, 20 μl of the oxidant diamide was added (final concentration of 20 mM), serving as fully oxidized sensor control. To a second well, 20 μl of the reductant DTT was added (final concentration of 100 mM), serving as fully reduced sensor control. Cells not expressing the roGFP2-based sensors, yet treated with diamide, DTT, or buffer, were used as blanks. The cells were pelleted by centrifugation at $20 \times g$ for 5 min at room temperature and placed in the instrument, kept at 30°C. A "steady state" was measured for 10 min, then the sample cells wells were subjected to experimental treatments adding 20 μl of a 10× experimental solution (e.g., containing $H_2O_2$), and the response followed for 120 min. Water was used as negative control for $H_2O_2$, and ethanol with final concentration 0.1% was used as negative control for antimycin A treatments. The degree of sensor oxidation (OxD) was calculated as in Equation (1) (Meyer & Dick, 2010; Morgan et al, 2011):

$$OxD = \frac{R^{\text{sample}} - R^{\text{RED}}}{\frac{I_{482}^{\text{OX}}}{I_{482}^{\text{RED}}} * (R^{\text{OX}} - R^{\text{sample}}) + (R^{\text{sample}} - R^{\text{RED}})} \quad \text{with} \quad R = \frac{I_{410}}{I_{482}} \quad (1)$$

$I_n$ = intensity at a given wavelength $n$; $I_{482}^{\text{OX}}$, and $I_{482}^{\text{RED}}$ = intensities at 482 nm upon complete oxidation by diamide or reduction by DTT. All experiments were performed in three independent biological replicates unless stated otherwise.

## "Acute stress-washout" experiment with genetically encoded sensors

Yeast cells were grown to mid-log phase in synthetic medium lacking the appropriate amino acids for plasmid selection. The cells were harvested by centrifugation at $1,500 \times g$ for 3 min at room temperature and resuspended in isosmotic buffer to a final concentration of 1.5 $OD_{600}$ units/ml. This process was repeated one more time. A bolus of water or $H_2O_2$ was added to bring the cell solution to the indicated final concentrations. After 10 min at 30°C, $H_2O_2$ was removed with two centrifugation steps at $1,500 \times g$ for 3 min at room temperature and following resuspension in isosmotic buffer to a final concentration of 1.5 $OD_{600}$ units/ml. Subsequently, each cell solution was transferred to a flat-bottomed 96-well imaging plate, and the experiment followed the classical fluorescence measurement of roGFP2-sensor oxidation.

## "Acute stress-washout" experiment with genetically encoded sensors and cytosolic translation inhibition experiment

Yeast cells were prepared as in the "Acute stress-washout" experiment until the removal of $H_2O_2$ with two centrifugation steps at $1,500 \times g$ for 3 min at room temperature. They were resuspended in the proper selective medium in the presence of either 50 μg/ml of the ribosome inhibitor cycloheximide (CHX) or dimethyl sulfoxide (DMSO) as control. After 2-h shaking at 30°C, cells were harvested and underwent two centrifugation steps at $1,500 \times g$ for 3 min at room temperature, followed by resuspension in isosmotic buffer to a final concentration of 1.5 OD600 units/ml. Subsequently, each cell solution was transferred to a flat-bottomed 96-well imaging plate,

and the experiment followed the classical fluorescence measurement of roGFP2-sensor oxidation. Controls at time zero, before the addition of cycloheximide or DMSO, but after the pre-treatment with $H_2O_2$, were also measured.

## Matrix DAO-based "Acute stress-washout" experiment with genetically encoded sensors

Yeast cells expressing Su9-DAO were grown to mid-log phase in synthetic medium lacking the appropriate amino acids for plasmid selection. The cells were harvested by centrifugation at $1,500 \times g$ for 3 min at room temperature and resuspended in pre-heated fresh medium to a final concentration of 1 $OD_{600}$ units/ml. This process was repeated one more time. A bolus of either D-Alanine or L-Alanine (final 0.15 M) was added to the cell suspension. The cells were kept shaking at 30°C for the indicated time. D-/L-Alanine was removed with two centrifugation steps at $1,500 \times g$ for 3 min at room temperature and following resuspension in isosmotic buffer to a final concentration of 1.5 $OD_{600}$ units/ml. Subsequently, each cell solution was transferred to a flat-bottomed 96-well imaging plate, and the experiment followed the classical fluorescence measurement of roGFP2-sensor oxidation.

## RNA preparation, sequencing, and analysis

Yeast cells were grown to early-log phase in synthetic medium lacking the appropriate amino acids for plasmid selection. 5 $OD_{600}$ units of cells were harvested by centrifugation at $1,500 \times g$ for 3 min at room temperature and resuspended in isosmotic buffer, and a second centrifugation step was necessary to retrieve pellets. For the Illumina library preparations, total RNA was first extracted from collected pellets using the RiboPure™ RNA Purification Kit, yeast (Invitrogen), according to the manufacturer's protocol. RNA-Seq libraries were prepared from total RNA using poly(A) enrichment of the mRNA (mRNA-Seq) and later analyzed on an Illumina HiSeq 4000 with a read-length of $2 \times 75$ base pairs. The sequencing data were uploaded to the Galaxy web platform (usegalaxy.org), and bioinformatics analysis was performed using the tools available on the public server (Afgan et al, 2016). Reads were aligned using the HISAT2 tool (Kim et al, 2015) and mapped using the S. cerevisiae R64-1-1.91 genome reference GTF file. The counting was performed using the featureCount tool. Statistical analysis was performed for three independent biological replicates, with the final differential expression between strains calculated using the package DESeq2 (Love et al, 2014). All tools were used with the default settings. The differential expression output data were represented in a volcano plot using the program OriginLab. The GO term analysis was performed using the online tool Gorilla (Eden et al, 2009).

## In vivo redox state analyses of Prx1 using mmPEG₂₄-based alkylation

Mature Prx1 without its MTS contains only one cysteine, the active-site cysteine at position 91 (C91). To distinguish the redox states of this cysteine, a mmPEG₂₄-based alkylation was applied, yielding a shift of ~ 2.4 kDa in the migration behavior of reduced Prx1 in a Western blot analysis (Kojer et al, 2012). Briefly, yeast cells were

grown to mid-log phase in synthetic medium lacking the appropriate amino acids for plasmid selection. Cells were harvested by centrifugation at 1,500 × g for 3 min at room temperature and resuspended in isosmotic buffer to a final concentration of 1.5 $OD_{600}$ units/ml. This process was repeated. Pellets of 2 $OD_{600}$ units were resuspended in 100 μl SDS-loading buffer containing either 10 mM $mmPEG_{24}$ (steady state), DMSO (unmodified), or 10 mM TCEP (maximally reduced) and boiled for 5 min at 96°C. The cells were then disrupted by vortexing with glass beads for 15 min in the dark. In the reduced sample, $mmPEG_{24}$ was added to a final concentration of 10 mM. After 40 min in the dark, all samples were analyzed in Western blots against Prx1.

Cells were also exposed to a pre-treatment as explained for the "Oxidation shock" experiment. In that case, after the pre-treatment 2 $OD_{600}$ units of cells were centrifuged at 20,000 × g for 30 s at room temperature and directly resuspended in 100 μl SDS-loading buffer containing 10 mM $mmPEG_{24}$. They were boiled, disrupted, and analyzed by Western blot as described above.

To test for irreversibility and the type of inactivation occurring in Prx1 after the hydrogen peroxide pre-treatment, 2 $OD_{600}$ units of cells were resuspended in 100 μl SDS-loading buffer containing either DMSO (unmodified), 10 mM TCEP (TCEP reduces disulfides and sulfenic acids only), or 10 mM $mmPEG_{24}$ (steady state), and boiled for 5 min at 96°C. The cells were then disrupted vortexing with glass beads for 15 min in the darkness. In the TCEP-reduced samples, $mmPEG_{24}$ was added to a final concentration of 10 mM. After 40 min in the darkness, all samples were analyzed in Western blots against Prx1.

To identify the redox state of Prx1 in the samples used in the *Matrix DAO-based* "Acute stress-washout" experiment, an aliquot of 2 $OD_{600}$ units of yeast for each time point was acquired at the step when D-/L-alanine was removed with two centrifugation steps at 1,500 × g for 3 min at room temperature and resuspended in isosmotic buffer to a final concentration of 1.5 $OD_{600}$ units/ml. The 2 $OD_{600}$ units of yeast were centrifuged at 20,000 × g for 30 s at room temperature and directly resuspended in 100 μl SDS-loading buffer containing 10 mM $mmPEG_{24}$. They were boiled, disrupted, and analyzed by Western blot as described above.

### "Acute stress" viability assay

Yeast cells were grown to mid-log phase in synthetic medium lacking the appropriate amino acids (S-medium) for plasmid selection or in YP-medium if no plasmid was present both with either 2% glucose or 2% galactose as carbon source. 10 $OD_{600}$ units of cells were harvested by centrifugation at 1,500 × g for 3 min at room temperature and then resuspended in water to a final concentration of 10 $OD_{600}$ units/ml. Subsequently, the cell solution was split into five tubes and centrifuged at 1,500 × g for 3 min at room temperature. The five 2 $OD_{600}$ pellets were then resuspended in 1 ml of either water or $H_2O_2$ to the final concentration. After shaking at 30°C for 30 min, 3.5 μl of cells solution was added to 25 ml water to a final concentration of approximately 0.00025 $OD_{600}$/ml. Next, 200 μl was plated on YPD plate (*ca.* 500 cells), and after growth for 5 days at 30°C, the number of viable cells was determined by the counting the colonies using the software *ImageJ*.

### "Halo" assay

Yeast cells were grown to mid-log phase in YP-medium with 2% glucose (YPD) as carbon source overnight and then diluted in YPD medium to a final concentration of 0.25 $OD_{600}$/ml. The culture was incubated at 30°C for 4 h and afterward diluted to a final concentration of 0.01 $OD_{600}$/ml in water. Subsequently, the suspension was used as inoculum and 100 μl spread on YP-medium 1% agar plates with either 2% glucose (YPD), 2% galactose (YPGal), or 2% glycerol (YPG). A 6-mm disk of cellulose loaded with 15 μl of 1 M diamide or 1 M $H_2O_2$ was placed at the center of the plate. From there, the reagent diffused into the plate. The plates were incubated at 30°C for 72 h, and then, the halo of growth around the disk was measured.

### Statistical analysis

In all figures, error bars represent mean ± standard deviation. *P*-values were determined using the *t*-test.

**Expanded View** for this article is available online.

### Acknowledgements

Research in the authors' laboratories is funded through grants of the German Research Council (DFG) to J.R. (SFB1218, TP B02, and SPP1710 RI2150/2-2) and B.M. (IRTG1830 and SPP1710 MO 2774/2-1). We thank Anja Wittmann (AG Riemer) for excellent technical help and generation of yeast strains. Cologne Center for Genomics (CCG) supported next-generation sequencing.

### Author contributions

GC, MD, BM, and JR designed the project, analyzed results, and wrote the article. GC, PSA, MNH, MM, TR, GPB, and EP performed the experiments and analyzed results.

### Conflict of interest

The authors declare that they have no conflict of interest.

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
