## [Review Process File · The EMBO Journal]

Hyperoxidation of mitochondrial peroxiredoxin limits H₂O₂-induced cell death in yeast

Gaetano Calabrese, Esra Peker, Prince Saforo Amponsah, Michaela Nicole Hoehne, Trine Riemer, Marie Mai, Gerd Patrick Bienert, Marcel Deponte, Bruce Morgan and Jan Riemer

Review timeline:

Submission date:	15th Jan 2019
Editorial Decision:	15th Feb 2019
Revision received:	17th May 2019
Editorial Decision:	27th Jun 2019
Revision received:	4th Jul 2019
Accepted:	8th Jul 2019

Editor: Elisabetta Argenzio

Transaction Report:

1st Editorial Decision

15th Feb 2019

Thank you for submitting your manuscript on the mechanism of H₂O₂-induced cells death to The EMBO Journal. Your study has been seen by three referees and their comments are enclosed below for your information.

As you can see, the referees find your work potentially of interest to the field of redox biology. However, they also raise several points that need to be addressed before they can support publication in The EMBO Journal. In particular, the referees request you to further investigate the role and physiological relevance of Prx1 hyperoxidation in H₂O₂-induced cell death.

Addressing these issues as suggested by the referees would be essential to warrant publication in The EMBO Journal. Given the overall interest of your study, I would like to invite you to submit a revised version of the manuscript according to the referees' requests. I should add that it is The EMBO Journal policy to allow only a single round of revision; acceptance of your manuscript will therefore depend on the completeness of your responses in this revised version.

REFeree REPORTS:

Referee #1:

General comments

In this paper, Riemer and colleagues address the compartmentation and fluxes of H₂O₂ between the yeast mitochondrial matrix and cytosol, using roGFP glutathione and peroxide sensors, concluding that cytosolic peroxiredoxins intercept the flux of exogenous H₂O₂ to the matrix and that H₂O₂ leads to the preferential oxidation of the matrix glutathione pool over the cytosolic one. In a second part, they identify the matrix peroxiredoxin, Prx1, as the enzyme that convey oxidizing equivalents to GSH, which undergoes inactivation by hyperoxidation at high doses H₂O₂, thereby preventing matrix glutathione oxidation. They then attempt to show a correlation between H₂O₂-induced matrix glutathione oxidation and cell death, using variants of Prxs with differing sensitivity to

hyperoxidation.

General comments

The first part of the paper deals with an interesting question already addressed in mammals (J Cell Biol, 2015, 211, 253), but the data provided remain incremental, some of which previously hinted to (preferential oxidation of matrix GSH relative to cytosol: FRBM, 2013, 65,436; cytosolic glutathione does not operate in H₂O₂ catabolism: JBC,2006, 281, 10420; importance of GSH over thioredoxin in the matrix, and differential oxidation of the cytosol and matrix GSH pool in response to H₂O₂: Free Radic Biol Med. 2016 May;94:55-65). In the second part, the proposed model is tantalizing, but still approximately supported by data, not only because of the highly correlative nature of experimental proofs, but also due to many inconsistencies between experiments and weakness of demonstrations, as explained below. Although the paper is cleverly organized, many experiments are not useful (Glr1 overexpression for instance), and confuse the reader. Important experimental details, such as medium choice, doses of H₂O₂ and rationale behind, are not easy to find or absent. The extremely high doses of H₂O₂ used to trigger Prx1 hyperoxidation (10-50 mM) profoundly question the physiological relevance of findings; heavier use of antimycin would have been appreciated.

In conclusion, this is a heavy study addressing important questions, and providing results, although incremental, of interest to the field of redox biology. However, authors push their model beyond realistic stances, based on the data they provide, which fully support the notion that Prx1 is the main mitochondrial matrix enzyme oxidizing GSH in this compartment, but are at this point insufficient to support the authors claim that Prx1 hyperoxidation limits H₂O₂-induced cell death by preventing GSH oxidation.

Major comments

1. Fig. 4 and 5

(i) Fig 4B shows that indeed there is a decrement of the roGFP response from 1 to 50 mM H₂O₂, but whether it is due to Prx1 hyperoxidation is contradicted by 4G that shows that Prx1 is fully hyperoxidized at 10 mM and in Fig S5C at 0.5 mM. Therefore, one cannot establish a cause and effect relationship between Prx1 hyperoxidation and the oxidation of GSH in these assays. The data of 4B and 2H are similarly discrepant. A reference H₂O₂-dose response of Prx1 hyperoxidation should be given, using MalPEG and WB.

(ii) If indeed Prx1 is fully oxidized at 0.5-1 mM H₂O₂, experiments that address the physiological relevance of this phenomenon should be done at this dose of H₂O₂, and Fig. 4D cannot be used as an indication of Prx1 hyperoxidation.

(iii) The result of 4F is unfortunately not at all in line with the model proposed by authors: if indeed Prx1 becomes hyperoxidized in the presence of high doses of H₂O₂, Wt cells should have the H₂O₂ phenotype of the deletion of Prx1 at high doses, but appears instead much more sensitive to acute stress than the Prx1 mutant.

2. Fig 5: Authors turned to gain-of-function mutations resistant to hyperoxidation, which is clever and has provided nice results (5B,C, D). Unfortunately, what shows fig 5D is that the hyperoxidation resistant mutant is just slightly more sensitive than Wt to acute H₂O₂ stress (4-5 % survival vs 9% at 25 mM H₂O₂), which again cast doubts on the authors' model that oxidized GSH is triggering cell death, not to consider the possibility that these experiments might be blurred by upregulation of CTT1.

3. Fig 6.

(i) The survival difference between the AOP Wt and mutant are again really not impressive, and not significant.

(ii) The only relevant variant is Prdx6, but data obtained cannot be interpreted because of the lack of information of its hyperoxidation status. The MalPEG-WB based assay should be used at the same H₂O₂ concentration that shows a difference in the acute stress assay between Prdx6 and Wt (10 and 25 mM).

4. Fig. 7. The effect of removing PRX1 in Dglr1 is very nice, clearly indicating major contribution of Prx1 to matrix GSH oxidation at baseline and in response to H₂O₂.

(i) However, the compared H₂O₂ sensitivity of Dprx1 and Dprx1Dglr1 should be tested to make the case that GSH oxidation promotes cell death. The Dglr1 cell proliferation lag phase and correction by deletion of PRX1 cannot be equated to the cell survival to toxic doses of H₂O₂.

- (ii) 7F is crucial but not informative as it stands: The level of GSH oxidation should be measured, parallel to the acute H₂O₂ challenge: 30 min incubation with 25 mM H₂O₂ should fully oxidize GSH in Wt and Wt expressing TPNO: as it stands, the experiment rather indicate that NADPH depletion is important in cell death.
- (iii) 7F experimental details are also totally lacking (see below).

Minor comments

1. Methods:

- (i) The cell growth stage and medium cell suspension dramatically influences H₂O₂ tolerance assays: at which O.D600 cells were taken for the acute stress experiments? Performing H₂O₂ tolerance assays with cells in the exponential phase of growth in culture media should allow decrease the doses of H₂O₂ at which cell dye.
- (ii) Why roGFP fluorescence measurements and acute stress wash out and viability were performed in iso-osmotic buffer, in which cells enters a stationary-like state, especially if the cell culture was saturated.
- (iii) The rationale behind switching from fermentative to respiratory or galactose media should be explained, and the media used when information missing indicated, the rational, if any, for switching the H₂O₂ doses over a very important range (0.1 to 50 mM).

2. Fig. 1:

- (i) The matrix H₂O₂ and GSH probes do not measure anything below 1 mM H₂O₂, a dose generally fungicidal in many laboratories (viability < 2%), which question the physiological relevance of data obtained at higher doses.
- (ii) Antimycin A triggers a strong oxidation of the H₂O₂ probe, but surprisingly, this does not translate into GSH oxidation: authors should explain or discuss this conundrum.
- (iii) The baseline degree of probe oxidation is different between G and H: explain.

3. Fig 2:

- (i) If indeed upregulation of Ctt1 intercepts exogenous H₂O₂ diffusing to the matrix, this should not apply to antimycin A-induced H₂O₂ production: is this the case? Is this explanation applying to the absence of a matrix H₂O₂ signal in the TRX3 mutant?
- (ii) Fig S2 C or D missing legend and description.

4. Fig. S3.

- (i) Asking "why is GSH more oxidized in the matrix than in the cytosol" is fine, but another maybe more relevant question would be why is GSH not oxidized in the cytosol?
- (ii) Why is the experiment of S3A done at 2.5 mM H₂O₂ instead of 1 mM as in Fig. 1?
- (iii) In the cytosol, as there is not oxidation of GSH, how would an increase of Glr1 have any effect? The proper cytosolic control should have been done with diamide that is known to oxidize GSH in all compartments.

5. Fig 3:

- (i) Why compare the prx1 mutant to the tsa1/tsa2 and por1 mutants? This might confuse the reader.
- (ii) If indeed CTT1 is upregulated in matrix redox enzyme mutants, it should be deleted to show that the no H₂O₂ response of the Trx2 and Trx3 mutants is only due to Ctt1 and that the no response of the Prx1 mutant to both Ctt1 upregulation and hyperoxidation. This is important because published data (Greetham et al., 2013, 18, 373, cited in the paper) indicate that Prx1 can be reduced by both GSH and Trx2.
- (iii) CTT1 upregulation is nicely shown to compensate for the lack of Prx1. Accordingly, does it compensate for hyperoxidized Prx1, which then could bias assay of H₂O₂ tolerance? For instance, the difference of H₂O₂ sensitivity between Dprx1 and Dprx1 expressing Prx1 or Prdx6 (Fig. 6A) might be a function of CTT1 expression, rather than GSH oxidation: this question should be addressed, at best by deletion of CTT1 in the respective strains used.

6. Fig. 4 and 5

- (i) 4D uses roGFP as control, but this probe equilibrates with the GSH/GSSG couple, and thus should correlate with the roGFP-Prx1 oxidation, if indeed only Prx1 oxidizes GSH. However, this is not what the data show.
- (ii) The MalPEG switch protocol is very confusing: One has to reduce lysates with DTT, better than TCEP and alkylate with AMS or MalPEG to appreciate the degree of hyperoxidation (see Nature,

2003, 425, 281). The use of arsenite confuses the reader: data are hard to see, not coherent and do not bring anything to the paper; Removing them would also allow suppressing Fig. 4 F, which actually carries errors: arrows 4 and 5: the disulfides should not become reduced by sodium arsenite. (iii) The experimental details are missing of how lysates are treated prior to MalPEG alkylation (5C). (vi) The cycloheximide experiment should have use the Prx1 alkylation protocol instead of the roGFP reporter.

7. Figure S3C is not described in the text.

8. Fig S4.

(i) 4D: why is the H₂O₂ phenotype only seen in YPGal and YPG and not in YPD?

(iii) 4G is not described in the text.

9. Fig. 6

The use of Tsa1, Prdx3 and Prdx5 is not relevant in the context of this paper, because these enzymes are reduced by thioredoxin, and not GSH. Furthermore, Tsa1 and Prdx3 should be fully oxidized, starting at 0.5 mM H₂O₂.

10. Fig.7

(i) Why a Dg1Dprx1 without ad-back cyto-Glr1 not tested?

(ii) Details of 7D should be given: experiment performed in the stationary phase, and mention the medium used (SD or Gal?). (iii) 7E: The proper control strain should be also grown in Gal medium and not SD. (iv) Are all cells pre-grown in Gal and which O.D. do they reach prior to the addition of H₂O₂?

Referee #2:

The manuscript by Calabrese et al. reports on a detailed analysis of the mechanism of H₂O₂-induced cell death in *Saccharomyces cerevisiae*. The experimental work has been performed - as far as I can judge, not being a yeast person myself - on a high standard. The manuscript still needs a bit of polishing, some of the issues are listed below.

Major issues:

The title should read: "Mitochondrial peroxiredoxin hyperoxidation limits

H₂O₂-induced cell death in yeast".

p. 15: The results of the experiment depicted in Fig. 5 E should be clearly stated in the Results section.

Minor issues:

p. 4, line 14 from bottom: Please re-phrase sentence (as H₂O₂ is not a redox couple).

p 17, line 8: we observed

p. 20, line 12: replace first "that" with "the"

p. 25, line 13: phenomenon

p. 26, line 8: reach

p. 26, line 6 from bottom: replace "has" with "have"

p. 30, line 4: insert "*Saccharomyces cerevisiae*" in front of "BY4742"

Referee #3:

Mitochondrial peroxiredoxin hyperoxidation limits H₂O₂-induced cell death

Calabrese et al

This manuscript aims to provide news insights into the mechanisms of H₂O₂ toxicity and concludes that mitochondrial Prx1p is involved in driving matrix glutathione oxidation and that this results in the observed toxicity. Oxidation of the active site Cys in Prx1p to a sulfenic/sulfonic acid acts as a

regulator to the system, switching off conversion of matrix GSH to GSSG and so limiting cell death under hyperoxidizing conditions.

Overall the manuscript is clear and well written, and the data probably supports the conclusions drawn. There are however issues with the manuscript which should be addressed. These primarily concern not what is present, but what is not. Specifically:

1) Page 3 para 2, the authors should provide evidence/citations regarding "the mediocre reactivity of H₂O₂ with most biological molecules"

2) Page 6. The authors rely heavily on GFP-based redox probes which have been modified to be more specific to thiol modifying redox species. The authors are careful in saying one has been modified "to increase the sensitivity of the probe to H₂O₂" and the "probe is predominantly reduced by endogenous GSH/glutaredoxins", but thereafter everything is black/white with no shades of grey. It would be very useful for some supplementary text to be added clarifying that these are not 100% specific the extent to which (if at all) this should be considered in data interpretation, what the possibly effects on "selectivity" are from modifying the mitochondrial redox enzymes etc.

3) For much of the data generated the authors consider primarily the maximal effect on the GFP-based probe. There appears to be more that the authors might derive from the data presented if they analyse the kinetics of oxidation and reduction they report. A simple example of this would be Figure 1C (1mM H₂O₂) where the small difference in maximal oxidation hides what appears to be minimal difference in oxidation rate, but a significant difference in the subsequent rate of reduction. Potentially more interesting differences may be present in other data shown.

4) It is essential that the authors re-examine the whole manuscript through the eyes of a non-yeast specialist. For example, the recipe for YPD and the synthetic media used need defining in the methods, figure labeling needs to be accessible to all readers e.g. in Fig S4D, YPD vs YPG media which contains galactose and which glucose (labelled here as D for dextrose as the norm in the yeast field, but not named as such in the main text) or Fig S4G where a non-specialist may struggle to work out that "Acute stress assay, Sgal" means that this was done in synthetic media with galactose as the carbon source (if it was galactose as the methods section implies only glucose was used in these assays)

5) It is unclear why different experiments were done in different media. For example, if no plasmid was present YPD media was used and if plasmid was present a synthetic media was used. This causes potential reader interpretation issues in places e.g. Figure S4F/G - is the massive difference in cell survival of the Dprx1 vs Dprx1+empty plasmid due to the empty plasmid or the media? & cross-comparison with Fig 5E becomes difficult as the media used in the assay apparently changes. What was the "fermentative media" used for the growth curves in Figure S2?

6) In the GFP-sensor assays "A steady state was measured for 10 minutes, then the sample wells were subjected to experimental treatments". Examining the figures there appears to be large differences in what is happening during this "steady state", with some being steady, but others showing significant increases during this time. How is this corrected for in calculating relative maximal changes? This for example impacts on Figure 6B.

7) Some of the over-expressed proteins were from yeast, others were non-codon optimized human proteins (PRXD3 cloning is missing from the methods section). What effect does this have on expression levels? Many proteins were tagged (but the tag not utilized), why? Is there any evidence this tagging may modulate function? Were they equally expressed? Were they successfully targeted to the mitochondria? Where Su9 presequence was added presumably any intrinsic N-terminal targeting was removed e.g. as stated in the methods section "genes of H.sapiens PRDX5 and PRDX6 were amplified..." which implies residue 1-214 of PRDX5, but was it just the mature protein region that was cloned? If so what was the start codon used for the mature protein? Were there any linkers between the protein and the C-terminal tags? Did the C91S or P233stop codon modulate protein expression levels (they are shown as separate blots) and if so could this modify interpretation of any of the results? For the Glr1 over-expression were both the mitochondrial and cytoplasmic forms equally over-expressed?

8) The antibody against Prx1 was generated in this study. No details of how the Prx1p was purified are given nor is there any data on specificity. Hopefully larger versions of the WB shown will be included in any resubmission as supplementary figures.

9) The majority of the work is based on yeast, but the authors show some HEK293 data in Figure S1B and then comment very briefly on the large differences in [H₂O₂] requirement to elicit a response. This is not picked up elsewhere. Perhaps either more explanation should be added or the figure cut.

10) The authors do not comment on why they observe a large effect of the water control on the apparent redox state of the matrix/cytosol e.g. figure 1G cytosol there is a large drop in signal in the Dtsa1Dtsa2 cells during the experimental time without addition of H₂O₂. This could potentially be combined with point 2 in as some supplementary text.

11) For the growth-curve experiments e.g. Figure S2D the authors state "cells exhibit impaired growth", but the doubling time appears to be unchanged (by eye), with the effect being in the lag-phase only. The authors should comment on this and any potential implications for the interpretation of other results.

12) For figure 3C (1mM) it is very difficult to differentiate between the curves. For this (and other figures) the authors should consider if it is essential to use the full 0->1 scale on the y-axis.

13) The authors say "observing a significant detrimental effect of PRX1 deletion for growth in the continuous presence of H₂O₂". From Figure S4D this is true in YP media using glycerol or galactose as a carbon source, but the opposite effect is seen when using glucose as a carbon source. this needs to be commented on and any implications discussed.

14) For figure S4D what is being shown? The radius as a % compared to what? The authors should clarify if they mean "halo of growth inhibition around the disk was measured" in the methods section.

15) For figure 4G the authors may want to comment on the relative intensities of the bands between samples e.g. for 1mM why in the steady state sample the sum of bands is much more intense than for the other lanes. In addition, the unmodified sample appears to migrate slightly slower in all cases than the TCP reduced lower band, why?

16) The legend for figure S4G is missing

17) Figure 5A, why is a modelled structure shown? The structure of Prx1p is in PDB (released October 2018).

18) Figure 5C, what is being examined, the steady state? if so this is not directly looking at hyperoxidation per se. The full assay done in figure 4G is required to eliminate between the possibilities. In addition, why in figure 4G is the ratio of the bands equal at 1mM H₂O₂, but in 5C the equivalent sample is >50% converted by 0.5mM H₂O₂? Are 4G and 5C representative WB or from a single experiment?

19) Figure S5A is a poor way of explaining the rationale behind the design. It needs more associated text. It should also be made clear that this will delete the terminal alpha-helix and 2 beta-strands. No explanation is given as to why this will make the protein more resistant to hyperoxidation.

20) There appear to be significant differences in the results shown in fig 4B (10mM) and in 5D left panel, why?

21) Since the water and H₂O₂ pre-states are different in some of the samples and the effects of the water control vary is the maximal H₂O₂ response a good measure to use in Figure 6B (the response to this could possibly be combined with points 2 and 10)

22) The raw data for H₂O₂ response for the empty plasmid is not shown in either 5D or S6B

Point-by-Point Response

Referee #1:

General comments

The first part of the paper deals with an interesting question already addressed in mammals (J Cell Biol, 2015, 211, 253), but the data provided remain incremental, some of which previously hinted to (preferential oxidation of matrix GSH relative to cytosol: FRBM, 2013, 65,436; cytosolic glutathione does not operate in H₂O₂ catabolism: JBC,2006, 281, 10420; importance of GSH over thioredoxin in the matrix, and differential oxidation of the cytosol and matrix GSH pool in response to H₂O₂: Free Radic Biol Med. 2016 May;94:55-65).

— We thank the referee for acknowledging the interest in the overall questions tackled in our study.

Nonetheless, we respectfully disagree with the assertion that the questions in our study have already been addressed in mammalian cells. In our opinion, our study focuses on completely novel aspects of H₂O₂ and glutathione regulation and reports novel insights and novel findings. Thus, we do not believe that our findings are incremental. We would contend that they are completely new.

— For more details regarding each of the studies suggested by the referee, please see our response below.

J Cell Biol, 2015, 211, 253:

This reference cited by the referee is an interesting study from the lab of David Ron. The paper uses the genetically encoded H₂O₂ sensor, HyPer, targeted to specific subcellular compartments, including the ER, mitochondrial matrix and cytosol, to investigate the role of ER-localized H₂O₂ in ER oxidative protein folding. As part of their array of control experiments, they also target catalase to each of the afore-mentioned subcellular compartments. They show that mitochondrial matrix HyPer is less responsive to exogenously applied H₂O₂ when a catalase is present in the matrix, compared to control cells. This is to be expected but we do not understand how it relates to our current study and do not understand the phrase 'the first part of the paper deals with an interesting question already addressed in mammals (J Cell Biol, 2015, 211, 253)'.

...preferential oxidation of matrix GSH relative to cytosol (FRBM, 2013, 65,436) AND

...differential oxidation of the cytosol and matrix GSH pool in response to H₂O₂: Free Radic Biol Med. 2016 May;94:55-65

In the case of this study, indeed, we do partially address this question and as the referee points out this aspect has previously been studied. However, even in this case we believe that we provide considerable new insights and technical advances over the cited studies. For example, in the cited study they only measure steady state roGFP2 oxidation, once, after 60 mins H₂O₂ treatment. In our study, we

— measure probe responses continuously over time, enabling us to gain a comprehensive picture of H₂O₂ dynamics (including oxidation and then recovery kinetics as well as precise determination of maximum oxidation of roGFPs). Furthermore, we are using Grx1-roGFP2 sensors, instead of an unfused roGFP2 protein as used in the cited study. Grx1-roGFP2 affords more rapid and specific equilibration with the glutathione pool compared to roGFP2. Finally, it is worth mentioning that the yeast strain employed in the cited study contains a mutation in Ybp1, which is a protein crucial for the function of the oxidative stress response Orp1-Yap1 pathway. In other words, these yeast strains are much more susceptible to oxidative challenges than most other yeast strains. Nonetheless, as a final note, even in these strains the authors had to use high (up to 10 mM) concentrations of H₂O₂ to induce probe responses and cell death. Thus, to address one of this referee's later comments, such high concentrations are frequently necessary in yeast studies and are not a peculiarity of our study.

...cytosolic glutathione does not operate in H₂O₂ catabolism (JBC,2006, 281, 10420):

We did not address the role of cytosolic glutathione in H₂O₂ metabolism in this study. We make no claim that cytosolic glutathione does not act in H₂O₂ catabolism.

— Although we do not draw attention to this issue, our results (and those of several previous studies) suggest that, in yeast at least, cytosolic glutathione can be oxidized in response to H₂O₂. Indeed, H₂O₂ is often used as a means of oxidizing cytosolic glutathione, for example Morgan et al, Nat. Chem. Biol., 2013 as well as in some of the references provided by the referee, i.e. Free Radic Biol Med. 2016 May;94:55-6 and FRBM, 2013, 65,436.

...importance of GSH over thioredoxin in the matrix, and differential oxidation of the cytosol and matrix GSH pool in response to H₂O₂ (Free Radic Biol Med. 2016 May;94:55-65):

In our study we only make the point that glutathione is involved in H₂O₂ reduction in the matrix. We show that for H₂O₂-induced GSH oxidation to occur there is a strict requirement for Prx1. This is an important insight, as there is considerable disagreement in the literature surrounding the role of glutathione in the reduction of Prx1 based on *in vitro* studies. Our results unambiguously demonstrate that glutathione is involved in the reduction of Prx1 inside living cells, although we cannot conclude whether glutathione is directly or indirectly involved in Prx1 reduction.

However, we do not (and indeed our data does not allow us to) claim that GSH is (or is not) more important than thioredoxin for reducing H₂O₂ in the matrix. The relative contribution of the two systems for matrix H₂O₂ detoxification remains unclear. It is even possible that there is considerable cross-talk between the thioredoxin and glutaredoxins systems in the matrix.

In conclusion, we feel that the suggestion that our study provides only incremental advances is incorrect. On the contrary, we make several new insights, each of which we believe will be of broad interest to the fields of redox biology and mitochondrial biology. Specifically:

- We show that Prx1 is exclusively required for H₂O₂-induced glutathione oxidation.
- We show that matrix glutathione is therefore involved in the reduction of matrix H₂O₂.
- We show that there is an adaptive response, in which cytosolic enzymes, principally Ctt1, are upregulated in response to Prx1 (and possibly other?) mitochondrial redox perturbations.
- The above point also indicates that, 1. Cells can sense and respond to matrix redox perturbations, although the molecular mechanism remains unknown. 2. Cytosolic redox enzymes, e.g. Ctt1, play an important role in regulating matrix H₂O₂ levels.
- We show that Prx1 hyperoxidation is protective against cell death, most likely due to minimizing oxidation of matrix glutathione.

To better highlight our novel insights we have reworked the discussion section.

In the second part, the proposed model is tantalizing, but still approximately supported by data, not only because of the highly correlative nature of experimental proofs, but also due to many inconsistencies between experiments and weakness of demonstrations, as explained below.

Below we address the criticism of the referee point-by-point.

Although the paper is cleverly organized, many experiments are not useful (Glr1 overexpression for instance) and confuse the reader.

We removed one set of experiments (as suggested by the referees) from the revised manuscript that was not necessary to convey the main message of our study: Data on E_{GSH} dynamics in human mitochondria (previous **Figure S1B**). We left the data concerning Glr1 in the manuscript as we feel they are important in the context of *GLR1* deletion experiments presented later in the manuscript.

Important experimental details, such as medium choice, doses of H₂O₂ and rationale behind, are not easy to find or absent.

We have extensively reworked figure legends, results text and the materials and methods section of all experiments. On the one hand, we now clearly indicate for each experiment the respective experimental conditions, including media choice. Additionally, we also explain rationale and experiment setup in a way that will be better understood by the “non-yeast” expert.

The extremely high doses of H₂O₂ used to trigger Prx1 hyperoxidation (10-50 mM) profoundly question the physiological relevance of findings; heavier use of antimycin would have been appreciated.

Whilst the doses of H₂O₂ used in this study might be seen as very high there are several important points to take into consideration that justify why such concentrations were used.

Viability assays were performed with 2 OD₆₀₀ units of cells, which equates to approximately 30,000,000 yeast cells, in a relatively small volume, 1 ml. Cell number and cell density are important variables in determining cell death following H₂O₂ treatment. The relatively high cell number and density used in our experiments mean that the total H₂O₂ ‘burden’ that is faced by each cell is relatively low. This is most obviously seen by the fact that cell viability, in many experiments, remains relatively high even after treatment with 10 mM exogenous H₂O₂. As the central question of our study was ‘What are the molecular mechanisms underlying H₂O₂ toxicity’ it was thus essential to use higher concentrations.

Another very important factor to be considered is the internal H₂O₂ concentration or, in other words, the gradient of H₂O₂ across the plasma membrane. Several studies have tried to address this question in different model organisms, yielding a wide-range of values, with internal H₂O₂ concentrations often reported as being several hundred times lower than exogenous H₂O₂ (Domenech et al, BMC Biol, 2018, 16: 61). We do not attempt to specifically provide further insight into this question, nonetheless, in response to this referee’s comment, we did address whether H₂O₂ diffusion across the plasma membrane is a limiting factor in determining H₂O₂ toxicity. We thereby provide evidence that high doses of H₂O₂ are necessary in part, because H₂O₂ only poorly traverses the plasma membrane of laboratory yeast strains as they lack functional aquaporins in the plasma membrane (Sabir et al, FEMS J 2017; Laize et al, Yeast 2000). Upon reintroduction of aquaporins, yeast cells also become more sensitive towards external H₂O₂ addition (**Supplementary Figure 4**).

We also provide additional experiments using a matrix-targeted D-amino acid oxidase to induce H₂O₂ production inside mitochondria upon addition of D-alanine (**novel Figure 4 and Supplementary Figure 4B, 4D and 4E**), which show that

Prx1 hyperoxidation can also be triggered in response to endogenous H₂O₂ production.

In conclusion, this is a heavy study addressing important questions, and providing results, although incremental, of interest to the field of redox biology. However, authors push their model beyond realistic stances, based on the data they provide, which fully support the notion that Prx1 is the main mitochondrial matrix enzyme oxidizing GSH in this compartment, but are at this point insufficient to support the authors claim that Prx1 hyperoxidation limits H₂O₂-induced cell death by preventing GSH oxidation.

— We appreciate that the referee acknowledges that Prx1 serves as main the “translator” between H₂O₂ and glutathione. We provide in the revised manuscript additional evidence for the link between Prx1-hyperoxidation, the resulting impairment of glutathione oxidation and the improved protection of yeast cells from cell death. We hope that we can convince this referee of the validity of our model based on this additional information.

Major comments

1. Fig. 4 and 5

— ***(i) Fig 4B shows that indeed there is a decrement of the roGFP response from 1 to 50 mM H₂O₂, but whether it is due to Prx1 hyperoxidation is contradicted by 4G that shows that Prx1 is fully hyperoxidized at 10 mM and in Fig S5C at 0.5 mM. Therefore, one cannot establish a cause and effect relationship between Prx1 hyperoxidation and the oxidation of GSH in these assays. The data of 4B and 2H are similarly discrepant. A reference H₂O₂-dose response of Prx1 hyperoxidation should be given, using MalPEG and WB.***

The referee highlights an important point; the apparent differences in the amounts of H₂O₂ that are required to either 1) elicit Prx1 hyperoxidation, (**Figures 4H, 5C and new Figure S6D**), 2) transfer oxidation through Prx1 onto roGFP2 in a fusion construct (**Figure 4E**), or 3) prevent oxidation transfer from H₂O₂ onto Grx1-roGFP2 (**Figure 4B, new Figure 4D**). We think that the reasons for these discrepancies are the very different setups of all three experiments (that we also extensively explain now in the materials and methods section and in the respective figure legends; see *below*). Collectively, the experiments clearly demonstrate that a certain threshold H₂O₂ concentration upon acute treatment decreases the E_{GSH} response, leads to blockage of the cysteine in Prx1 and blocks the capacity of H₂O₂ to oxidize a roGFP2 that is fused to roGFP2.

The experiment presented in **Figure 4B** has been done in the following way: Cells were exposed to increasing concentrations of H₂O₂, then the H₂O₂ was washed away, the cells were pelleted into a 96 well plate, and then the E_{GSH} response was assessed using addition of 1 mM H₂O₂. The time between the two H₂O₂ exposures

was thereby approximately 5 minutes. Conversely, for the maleimide shift experiments depicted in **Figures 4H, 5C and S6D-F**, cells were treated with increasing amounts of H₂O₂ and then the redox state was frozen by addition of TCA. Thus, the discrepancy with respect to the required amount of H₂O₂ between these experiments might stem from the recovery time that the cells in the recovery experiment have.

We also provide two additional experiments to strengthen the link between acute H₂O₂ treatment – Prx1 inactivation and concomitant loss of oxidation transfer from H₂O₂ onto glutathione: 1) an *in vitro* reconstitution experiment (for the referees, see below), 2) an experiment using D-amino acid oxidase as H₂O₂ donor in the mitochondrial matrix (**new Figure 4D**).

The *in vitro* experiment resembles our *in vivo* setup in that we assess Grx1-roGFP2 dynamics in a Prx1-glutathione context. Specifically, we tested the capacity of Prx1 and H₂O₂-pretreated Prx1 to transfer oxidation from increasing amounts of H₂O₂ onto a glutathione buffer, and read it out using Grx1-roGFP2. Using this experiment, we can show that Prx1 indeed oxidizes the Grx1-roGFP2 probe in the presence of glutathione. Conversely, H₂O₂-pretreated Prx1 did so to a far lesser extent. We provide this experiment only to the referees as we feel that this completely new experimental setup and read-out would complicate the manuscript. If requested by the referees, we are happy to include the experiment in the revised version.

We also extended our engineering approaches and targeted D-amino acid oxidase to the mitochondrial matrix. In the presence of D-alanine but not L-alanine, this enzyme locally produces H₂O₂. With this tool, we can show that upon preincubation with D-alanine, Grx1-roGFP2 responds to a lesser extent to external H₂O₂ addition than if not incubated with D-alanine or if incubated with L-alanine (**compare novel Figures 4D and Supplementary Figure 4B**). In line with this observation, the active-site cysteine in Prx1 becomes blocked under these conditions (**novel Supplementary Figure 4D and 4E**).

ii) If indeed Prx1 is fully oxidized at 0.5-1 mM H₂O₂, experiments that address the physiological relevance of this phenomenon should be done at this dose of H₂O₂, and Fig. 4D cannot be used as an indication of Prx1 hyperoxidation.

Please see our comment to the point above. The varying experimental setups necessitate the use of different H₂O₂ concentrations for different assays. Therefore, although as the referee rightly suggests all experiments should be performed with the same amount of H₂O₂ this is in practice unfortunately not possible.

(iii) The result of 4F is unfortunately not at all in line with the model proposed by authors: if indeed Prx1 becomes hyperoxidized in the presence of high doses of H₂O₂, Wt cells should have the H₂O₂ phenotype of the deletion of Prx1 at high doses, but appears instead much more sensitive to acute stress than the Prx1 mutant.

We assume the referee is referring to the acute stress assay presented in **Figure S4**? Indeed, wild type cells are more sensitive to H₂O₂ than the *PRX1* deletion. However, this is to be expected as cells initially (*i.e.* before inactivation of Prx1) contain a fully active Prx1 that oxidizes glutathione. This also becomes apparent with the P233STOP variant: although even this variant is inactivated at very H₂O₂ concentrations, it still is slightly more sensitive in the acute stress assay compared to the wild-type.

2. Fig 5: Authors turned to gain-of-function mutations resistant to hyperoxidation, which is clever and has provided nice results (5B,C,D). Unfortunately, what shows fig 5D is that the hyperoxidation resistant mutant is just slightly more sensitive than Wt to acute H₂O₂ stress (4-5 % survival vs 9% at 25 mM H₂O₂), which again cast doubts on the authors' model that oxidized GSH is triggering cell death, not to consider the possibility that these experiments might be blurred by upregulation of CTT1.

We respectfully disagree with the referee's comment in this instance. We would assert that the hyperoxidation-resistant mutant (P233STOP) is clearly more sensitive than wild-type Prx1 to acute H₂O₂ stress. The presentation of acute stress assays is with a logarithmic survival percentage scale. At 10 mM H₂O₂ for example, the mean wild-type survival is 12.5% vs. 31.5% survival of the P233STOP mutant, a ~2.5-fold difference.

3. Fig 6. (i) The survival difference between the AOP Wt and mutant are again really not impressive, and not significant. (ii) The only relevant variant is Prdx6, but data obtained cannot be interpreted because of the lack of information of its hyperoxidation status. The MalPEG-WB based assay should be used at the

same H₂O₂ concentration that shows a difference in the acute stress assay between Prdx6 and Wt (10 and 25 mM).

The referee is correct that there is not a statistically significant difference between viability of strains expressing PfAOPwt and PfAOPL109M. However, we argue that there is a clear trend in the expected direction between these mutants (for example, AOP L109M vs WT; 9.5% survival vs 22.9% at 10 mM H₂O₂). Furthermore, we feel that these two mutants should also be seen in the context of all the other peroxiredoxin variants tested in this experiment, all of which conform to the expected trend.

The sensitivity of the PRDX6 variant in the acute stress assay is clearly different compared to WT-Prx1 and $\Delta prx1$ (**Figure 6A**). In the revised manuscript, we provide information on the redox state of matrix-targeted PRDX6 upon acute H₂O₂ treatment (**new Supplementary Figure 6F**). We observe that PRDX6 cysteines become inaccessible to mmPEG at low external H₂O₂ concentrations. However, this can be reverted by addition of TCEP, which indicates that PRDX6 cysteines were not hyperoxidized to a significant extent even at 10 mM external H₂O₂. Conversely, for Prx1 no reversal can be achieved indicating hyperoxidation of the Prx1 cysteine at low H₂O₂ concentrations (**new Supplementary Figure 6D**).

4. Fig. 7. The effect of removing PRX1 in Dglr1 is very nice, clearly indicating major contribution of Prx1 to matrix GSH oxidation at baseline and in response to H₂O₂. (i) However, the compared H₂O₂ sensitivity of Dprx1 and Dprx1Dglr1 should be tested to make the case that GSH oxidation promotes cell death. The Dglr1 cell proliferation lag phase and correction by deletion of PRX1 cannot be equated to the cell survival to toxic doses of H₂O₂.

We thank the referee for this excellent suggestion. We performed the respective experiment comparing WT, $\Delta prx1$, $\Delta glr1$ and $\Delta glr1 \Delta prx1$. We thereby find that while $\Delta glr1$ cells are more sensitive to acute H₂O₂ stress compared to wild type and $\Delta prx1$ cells, additional deletion of PRX1 in this background ($\Delta prx1 \Delta glr1$) partially rescues this phenotype (**new Figures 7C and 7D**).

As a consequence of this clear observation, we removed the former TPNOX-related panels from **Figure 7** from the manuscript (see Figure below). However, if the referees think they should be reintroduced we are happy to do so.

(ii) 7F is crucial but not informative as it stands: The level of GSH oxidation should be measured, parallel to the acute H₂O₂ challenge: 30 min incubation with 25 mM H₂O₂ should fully oxidize GSH in Wt and Wt expressing TPNO: as it stands, the experiment rather indicate that NADPH depletion is important in cell death.

As explained in the previous point, we decided to take these data out of the manuscript. We now provide a different data set using *GLR1* and *PRX1* deletion strains to test the influence of NADPH and GSSG on cell survival (**novel Figures 7C and D**).

At 25 mM external H₂O₂, the matrix Grx1-roGFP2 probe is indeed fully oxidized. However, it is important to point out that the probe can only measure over a limited range of glutathione oxidation, approx. -320mV - -240 mV. Thus, full oxidation of the probe does not equate full oxidation of the glutathione pool, far from it. Furthermore, previous data from cytosolic measurements, (e.g. see data in Morgan et al Nat Chem Biol 2013, show that even high amounts of H₂O₂ (up to 20 mM) did not result in full oxidation of cytosolic glutathione probe). Here, we provide data that recovery of the glutathione pool after acute H₂O₂ treatment takes place very rapidly (washout takes ca 3-5 min). In **Figure 4B**, the steady states of the matrix-targeted Grx1-roGFP2 probe are already relatively reduced again even after pretreatment with up to 50 mM H₂O₂ (water pretreatment OxD = 0.12, 50 mM H₂O₂ pretreatment OxD = 0.3).

(iii) 7F experimental details are also totally lacking (see below).

We have improved experimental descriptions throughout the text, the figure legends and the materials and methods section. As described above, we have now removed old **Figure 7F** due to the addition of new experimental data.

Minor comments

1. Methods: (i) The cell growth stage and medium cell suspension dramatically influences H₂O₂ tolerance assays: at which O.D600 cells were taken for the acute stress experiments? Performing H₂O₂ tolerance assays with cells in the exponential phase of growth in culture media should allow decrease the doses of H₂O₂ at which cell dye.

Nearly all experiments were performed at an OD₆₀₀ between 1–1.5. As described, we have considerably amended relevant sections of the manuscript to clarify the details.

— **(ii) Why roGFP fluorescence measurements and acute stress wash out and viability were performed in iso-osmotic buffer, in which cells enters a stationary-like state, especially if the cell culture was saturated.**

As described above, in nearly all experiments the cells were harvested at early-to-mid exponential phase.

— Indeed, we do perform our experiments in an iso-osmotic buffer. This is a standard protocol that we have developed over the past years working with these sensors in yeast. We (and others) have observed that the precise composition of the cell suspension buffer (or growth media) can have a significant impact on the response of cytosolic and mitochondrial roGFP2-based probes responses. Even small changes in glucose concentration or ionic composition can lead to considerably different probe response. The prevailing opinion amongst researchers working closely in this area is that these differences are reflective of real biological differences in either a) how readily H₂O₂ crosses the plasma membrane and enters the cells: possibly due to osmotic or membrane potential changes, and/or b) the cellular capacity to detoxify the incoming H₂O₂. However, although known for several years a precise understanding of these effects remains lacking. In the absence of any satisfactory understanding, we choose to use a standardized buffer to minimize variability between experiments. An additional advantage of using a standardized buffer is that it minimizes fluorescence background, for example compared to imaging in growth media.

Nonetheless, it is important to point out that in the acute stress assays, the iso-osmotic buffer is only used during the acute stress phase. After H₂O₂ washout, cells are plated onto rich medium-containing agar plates.

(iii) The rationale behind switching from fermentative to respiratory or galactose media should be explained, and the media used when information missing indicated, the rationale, if any, for switching the H₂O₂ doses over a very important range (0.1 to 50 mM).

Nearly all experiments are performed in galactose-containing media, as galactose does not suppress synthesis of mitochondrial proteins and thereby mitochondrial biogenesis (compared to glucose, “glucose repression”). In some experiments, we had to deviate from this, e.g. TPNOX is under the control of a GAL promotor and accordingly to repress or induce this enzyme we need to grow our cells in glucose or galactose, respectively. For the recovery and analysis of cells after the acute stress assay, cells are always plated onto glucose-containing medium, as this is the preferred carbon source of yeast. It provides optimal cell growth conditions and prevents further cell selection based on metabolic conditions.

Throughout the manuscript, different H₂O₂ concentrations are used. These H₂O₂ concentrations were always established in initial titration experiments (e.g. **Figures 1C,D** for the sensors). For many experiments, we have collected data at a range of different H₂O₂ concentrations in addition to those presented in the manuscript. However, for the sake of clarity of the manuscript, we chose to provide data with specific H₂O₂ concentrations.

We now more extensively explain the rationale between changes in the employed media and the use of different H₂O₂ concentrations throughout the text and in the figure legends.

2. Fig. 1: (i) The matrix H₂O₂ and GSH probes do not measure anything below 1 mM H₂O₂, a dose generally fungicidal in many laboratories (viability < 2%), which question the physiological relevance of data obtained at higher doses.

Consistent with our answer above, the yeast response to H₂O₂ is dependent upon a number of factors including cell number and cell density and also shows variability between strains. The lack of response of the sensors has been shown to be reflective of the biological reality that cellular redox species in yeast are tightly regulated, for example, Morgan et al, 2013, Nat. Chem. Biol and Morgan et al, 2016, Nat. Chem. Biol.

Although responses may be minimal or absent in wild-type, they can readily be observed for example in mutant cells, for example the glutathione pool is considerably more sensitive to perturbation in $\Delta glr1$ cells, whilst matrix H₂O₂ and glutathione are more sensitive in $\Delta tsa1\Delta tsa2$ cells, supporting the notion that the cytosolic peroxiredoxins considerably limit the amount of exogenous H₂O₂ reaching the mitochondrial matrix.

Indeed, as described above, such high concentrations of H_2O_2 are necessary and are used in studies in many yeast laboratories, including in some of the references cited by this referee, for example *Free Radic Biol Med.* 2016 May;94:55-65.

(ii) Antimycin A triggers a strong oxidation of the H_2O_2 probe, but surprisingly, this does not translate into GSH oxidation: authors should explain or discuss this conundrum.

Antimycin A indeed leads to oxidation of roGFP2-Tsa2 Δ C_R, however with very different kinetics (more delayed response) compared to addition of external H_2O_2 . It does indeed not lead to a strong increase of Grx1-roGFP2 oxidation. Unfortunately, we do not have a clear mechanistic explanation for this observation, although we can speculate on possible reasons, see below.

On a more speculative note we would like to point out that upon external H_2O_2 addition, the Grx1-roGFP2 probe signal recovers with time, although using chemical assays, we still detect H_2O_2 extracellularly. This might suggest that cells respond rapidly to H_2O_2 stress, e.g. by increasing the production of NADPH or preventing H_2O_2 membrane diffusion by inhibiting transporters, perhaps such adaptor responses can explain the resistance of the glutathione pool to antimycin A-induced H_2O_2 .

(iii) The baseline degree of probe oxidation is different between G and H: explain.

We are uncertain what the referee is referring to in this comment. We feel that the baselines, i.e. before addition of H_2O_2 between **Figure 1G** and **Figure 1H** are exactly the same. In both cases they start at the same OxD and decrease over time very similarly, consistent with the consumption of oxygen in the assay, as has been described previously (Morgan et al, *Nat. Chem. Biol.*, 2016).

3. Fig 2: (i) If indeed upregulation of Ctt1 intercepts exogenous H_2O_2 diffusing to the matrix, this should not apply to antimycin A-induced H_2O_2 production: (a) is this the case? (b) Is this explanation applying to the absence of a matrix H_2O_2 signal in the TRX3 mutant?

(a) This is a good point. We performed experiments in this direction and could indeed confirm that this is the case. We now added the data as new **Figure S2E**.

(b) We believe that this is the case although we have not specifically checked that this is the case for Trx3. Instead, we focused on the relationship between Prx1 and Ctt1. However, we contend that the adaptive response is likely to be more generally induced upon perturbation of mitochondrial redox enzymes.

(ii) Fig S2 C or D missing legend and description.

We added the legend and description.

4. Fig. S3. (i) Asking "why is GSH more oxidized in the matrix than in the cytosol" is fine, but another maybe more relevant question would be why is GSH not oxidized in the cytosol?

This is indeed an interesting question (although in our manuscript we focus on E_{GSH} in the matrix).

— GSH is indeed oxidized in the cytosol upon addition of external H_2O_2 and this depends on the presence of Tsa1/2 (see *Figure below*, although our data do not allow to judge whether this is a direct or indirect effect). Different from the matrix, the cytosol commands additional ways to maintain a reduced E_{GSH} . This includes export of GSSG from the cytosol (Minich et al, J. Neurochem 2006; Morgan et al, Nat. Chem. Biol, 2013), but also the presence of NADPH-independent modes of H_2O_2 handling (catalases) which prevents crosstalk with GSSG reduction that also depends on NADPH. Additionally, the supply of NADPH in the cytosol by the pentose phosphate pathway might be more efficient than NADPH-providing processes in the matrix. The combination of these effects and the (likely) more efficient oxidation of GSSG by Prx1 compared to Tsa1/2 might explain the differences in E_{GSH} response upon H_2O_2 treatment.

—

(ii) Why is the experiment of S3A done at 2.5 mM H₂O₂ instead of 1 mM as in Fig. 1?

This experiment was also performed at an H_2O_2 concentration of 1 mM. This data is shown in **Figure 7A**.

(iii) In the cytosol, as there is not oxidation of GSH, how would an increase of Glr1 have any effect? The proper cytosolic control should have been done with diamide that is known to oxidize GSH in all compartments.

As indicated above, there is H₂O₂-dependent oxidation of GSH.

5. Fig 3: (i) Why compare the prx1 mutant to the tsa1/tsa2 and por1 mutants? This might confuse the reader.

The two mentioned deletion strains were used already in **Figure 1** to understand the impact of the cytosol and membrane transport on H₂O₂ dynamics in the mitochondrial matrix. We thereby concluded that Tsa1 and Tsa2 protect the matrix from exogenous H₂O₂. In line with this, deletion of *TSA1* and *TSA2* result in increased deflection of matrix E_{GSH} upon H₂O₂ addition. Por1 facilitates H₂O₂ transport over the outer mitochondrial membrane. In line with this, E_{GSH} exhibits a smaller response in a *POR1* deletion. Thus, both strains confirm the correlation between matrix H₂O₂ levels and E_{GSH} dynamics. Understanding this link is the main point of this figure and we would thus prefer to keep these panels in.

(ii) If indeed CTT1 is upregulated in matrix redox enzyme mutants, it should be deleted to show that the no H₂O₂ response of the Trr2 and Trx3 mutants is only due to Ctt1 and that the no response of the Prx1 mutant to both Ctt1 upregulation and hyperoxidation. This is important because published data (Greetham et al., 2013, 18, 373, cited in the paper) indicate that Prx1 can be reduced by both GSH and Trx2.

In this case, we present the data with the $\Delta trx3$ and $\Delta trr2$ strains for the sake of completeness. For a more detailed understanding, we chose to focus specifically on the relationship between Ctt1 and Prx1.

Whilst the referee is correct that the relationship between other mitochondrial redox enzymes and Ctt1 (the adaptive response) would be interesting to study, we feel that such a comprehensive study is well beyond the scope of this study and would represent at least one publication in its own right.

(iii) CTT1 upregulation is nicely shown to compensate for the lack of Prx1. Accordingly, does it compensate for hyperoxidized Prx1, which then could bias assay of H₂O₂ tolerance? For instance, the difference of H₂O₂ sensitivity between Dprx1 and Dprx1 expressing Prx1 or Prdx6 (Fig. 6A) might be a function of CTT1 expression, rather than GSH oxidation: this question should be addressed, at best by deletion of CTT1 in the respective strains used.

This is an important point that we have addressed by analyzing a strain lacking PRX1 with a strain lacking both PRX1 and CTT1 (new Figures S4N). We thereby find that cells lacking Ctt1 alone perform worse in an acute stress assay than cells lacking both, PRX1 and CTT1. This indicates that even in the absence of the main player of the adaptive cytosolic response, the absence of PRX1 is still protective. Thus, we think that Ctt1 level increase and Prx1 hyperoxidation are synergistic strategies to maintain a reduced matrix E_{GSH} .

6. Fig. 4 and 5 (i) 4D uses roGFP as control, but this probe equilibrates with the GSH/GSSG couple, and thus should correlate with the roGFP-Prx1 oxidation, if indeed only Prx1 oxidizes GSH. However, this is not what the data show.

Figure 4E (old 4D) compares the roGFP2 sensor with roGFP2-Prx1.

Indeed, the roGFP2 sensor equilibrates with E_{GSH} using local glutaredoxins (in this case Grx2 as we measure in the matrix). When we compare **Figure 4E** with **Figure 1D** (use of matrix-targeted Grx1-roGFP2), then we observe pretty similar behavior of the probes (compare reaction to 1 mM H₂O₂) with roGFP2 alone being slightly less sensitive as expected (different kinetics between fused and unfused probes and limiting amounts of local glutaredoxins).

Prx1 directly interacts with H₂O₂ and transfers oxidation onto roGFP2, i.e. roGFP2 is the direct reductant of Prx1. Thus in the context of the roGFP2-Prx1 probe, the roGFP2 redox state will be determined by both Prx1-mediated oxidation and Grx2 mediated equilibration with the glutathione pool (Morgan et al, Nature Chem Biol 2016). Thus only if the Prx1 in the roGFP2-Prx1 probe becomes completely hyperoxidized would we expect the same behavior for roGFP2 and Prx1-roGFP2

(ii) The MalPEG switch protocol is very confusing: One has to reduce lysates with DTT, better than TCEP and alkylate with AMS or MalPEG to appreciate the degree of hyperoxidation (see Nature, 2003, 425, 281). The use of arsenite confuses the reader: data are hard to see, not coherent and do not bring anything to the paper; Removing them would also allow suppressing Fig. 4 F, which actually carries errors: arrows 4 and 5: the disulfides should not become reduced by sodium arsenite.

(iii) The experimental details are missing of how lysates are treated prior to MalPEG alkylation (5C).

We realized from the responses of the referees that the presentation of the maleimide shift assay was confusing. We thus reworked what is now **Figure 4F** and the presentation of the maleimide shift assay. We also made the respective changes in the text and figure legends to better explain the assay and the employed conditions.

Our assay is based on changes in the migration pattern of Prx1 as induced by modification of the active site cysteine with maleimide. Modification will result in a slower migration of the protein on SDS-PAGE and is dependent on the presence of the active site cysteine in the reduced state (-SH). If the cysteine thiol is blocked (e.g. -SSG, -SOH, -SO₂H), then the maleimide will not modify it. Treatment with the reductant TCEP (that is not well reactive with maleimides) prior to maleimide modification will reveal whether the cysteine is blocked in a TCEP-reversible way (this reversibility is true e.g. for -SSG or -SOH, Reisz et al, FEBS J 2013). Thus, pretreatment with TCEP and comparing to the directly modified sample reveals whether the protein had been hyperoxidized or not. In this case, the migration behavior will not change upon TCEP pretreatment. A good example is **Figure S6F**: Pretreatment with TCEP frees the PRDX6 cysteines and indicates that the protein is not hyperoxidized even at high H₂O₂ concentrations.

7. Figure S3C is not described in the text.

We moved this panel now into Figure 3 as new panel **3C**. We think this improves the flow of the manuscript. Consequently, this Figure is now also described in the text and the figure legend.

8. Fig S4. (i) 4D: why is the H₂O₂ phenotype only seen in YPGal and YPG and not in YPD? (iii) 4G is not described in the text.

On glucose, yeast cells contain very few mitochondria and these are metabolically not very active. This glucose repression is released upon growth on galactose and glycerol. Thus, we would expect that differences between $\Delta prx1$ (matrix enzyme) and wildtype only become apparent on medium relying on the active use of mitochondria.

9. Fig. 6 The use of Tsa1, Prdx3 and Prdx5 is not relevant in the context of this paper, because these enzymes are reduced by thioredoxin, and not GSH. Furthermore, Tsa1 and Prdx3 should be fully oxidized, starting at 0.5 mM H₂O₂.

We respectfully disagree with the referee as these enzymes in the context of the intact cell lead to the oxidation of glutathione (**Figure S6B**). We cannot conclude whether the oxidation of glutathione is direct or indirect but the data nonetheless complement our interpretation that glutathione oxidation directly correlates with cell death. It is worth pointing out that recent studies have shown that glutathione/glutaredoxins can reduce typical 2-Cys peroxiredoxins (e.g. Peskin et al, J. Biol. Chem. (2016) 5:291(6):3053-62).

— The referee's assertion that Tsa1 and PRDX3 should be hyperoxidized by 0.5 mM H₂O₂ is also not correct. To elaborate: *in vitro* the referee would certainly be correct. However, what is important in our assays is the intracellular H₂O₂ concentration. This is less clear. A recent study in the fission yeast *S. pombe*, reported an H₂O₂ gradient across the plasma membrane of ~300:1 (Domenech et al, BMC Biol (2018)). If also applicable to budding yeast, then 0.5 mM exogenous H₂O₂ would equate to 1.6 μM in the cytosol. The concentration in the mitochondrial matrix would be expected to be even lower, possibly much lower.

— **10. Fig.7**

(i) Why a Dglr1Dprx1 without ad-back cyto-Glr1 not tested?

We tested this and included the data set in **Figure 7A**. Using matrix Grx1-roGFP2, the difference in response between $\Delta prx1\Delta glr1$ and $\Delta prx1\Delta glr1$ +cyto-Glr1 is very small.

(ii) Details of 7D should be given: experiment performed in the stationary phase and mention the medium used (SD or Gal?).

As requested by all referees, we improved the description of experimental settings throughout the manuscript.

(iii) 7E: The proper control strain should be also grown in Gal medium and not SD.

This is in principle correct. However, TPNOX is under the control of a GAL promotor (induction on galactose, repression on glucose). Thus, strains can be compared among each other on the same carbon source.

Note: as state above, we removed the TPNOX data set from the manuscript.

(iv) Are all cells pre-grown in Gal and which O.D. do they reach prior to the addition of H₂O₂?

All strains are normally grown in galactose-containing media to exponential phase and a similar OD₆₀₀ between 1 and 1.5. The exceptions (e.g. growth on glucose for repression of *GAL* promotor) are indicated.

Referee #2:

The manuscript by Calabrese et al. reports on a detailed analysis of the mechanism of H₂O₂-induced cell death in Saccharomyces cerevisiae. The experimental work has been performed - as far as I can judge, not being a yeast person myself - on a high standard. The manuscript still needs a bit of polishing, some of the issues are listed below.

We thank the referee for the positive evaluation of the experimental standards of our experiments.

Major issues:

The title should read: "Mitochondrial peroxiredoxin hyperoxidation limits H₂O₂-induced cell death in yeast".

We changed the title according to the suggestion of the referee.

p. 15: The results of the experiment depicted in Fig. 5 E should be clearly stated in the Results section.

As also requested by the other referees, we extended our experimental descriptions throughout the manuscript and the figure legends.

Minor issues:

p. 4, line 14 from bottom: Please re-phrase sentence (as H₂O₂ is not a redox couple).

p 17, line 8: we observed

p. 20, line 12: replace first "that" with "the"

p. 25, line 13: phenomenon

p. 26, line 8: reach

p. 26, line 6 from bottom: replace "has" with "have"

p. 30, line 4: insert "Saccharomyces cerevisiae" in front of "BY4742"

We corrected these mistakes.

Referee #3:

This manuscript aims to provide news insights into the mechanisms of H₂O₂ toxicity and concludes that mitochondrial Prx1p is involved in driving matrix glutathione oxidation and that this results in the observed toxicity. Oxidation of the active site Cys in Prx1p to a sulfinic/sulfonic acid acts as a regulator to the system, switching off conversion of matrix GSH to GSSG and so limiting cell death under hyperoxidizing conditions.

Overall, the manuscript is clear and well written, and the data probably supports the conclusions drawn. There are however issues with the manuscript which should be addressed. These primarily concern not what is present, but what is not.

Specifically:

1) Page 3 para 2, the authors should provide evidence/citations regarding "the mediocre reactivity of H₂O₂ with most biological molecules".

We have now added appropriate citations to support this statement.

2) Page 6. The authors rely heavily on GFP-based redox probes which have been modified to be more specific to thiol modifying redox species. The authors are careful in saying one has been modified "to increase the sensitivity of the probe to H₂O₂" and the "probe is predominantly reduced by endogenous GSH/glutaredoxins", but thereafter everything is black/white with no shades of grey. It would be very useful for some supplementary text to be added clarifying that these are not 100% specific the extent to which (if at all) this should be consider in data interpretation, what the possibly effects on "selectivity" are from modifying the mitochondrial redox enzymes etc.

We agree with the referee that every experimental tool must be employed with great care and the resulting data should be interpreted cautiously. We did this in the previous version of the manuscript as the referee acknowledges. In the revised manuscript version we point the reader to a review that in detail discusses the mechanisms and pitfalls of the roGFP2-based probes (p.6 las line).

3) For much of the data generated the authors consider primarily the maximal effect on the GFP-based probe. There appears to be more that the authors might derive from the data presented if they analyse the kinetics of oxidation and reduction they report. A simple example of this would be Figure 1C (1mM H₂O₂) where the small difference in maximal oxidation hides what appears to be minimal difference in oxidation rate, but a significant difference in the subsequent rate of reduction. Potentially more interesting differences may be present in other data shown.

We agree with the referee that different parameters of the curve (e.g. max OxD, slopes and width of peak) are important for full probe interpretation. We always show full readouts and point the reader at times to specific differences. However, in the interest of keeping the already very long manuscript at a reasonable length, we decided not to describe all curves in full detail.

4) It is essential that the authors re-examine the whole manuscript through the eyes of a non-yeast specialist. For example, the recipe for YPD and the synthetic media used need defining in the methods, figure labeling needs to be accessible to all readers e.g. in Fig S4D, YPD vs YPG media which contains galactose and which glucose (labelled here as D for dextrose as the norm in the yeast field, but not named as such in the main text) or Fig S4G where a non-specialist may struggle to work out that "Acute stress assay, Sgal" means that this was done in synthetic media with galactose as the carbon source (if it was galactose as the methods section implies only glucose was used in these assays)

We thank the referee for this advice and worked on all parts of the manuscript to render it more accessible to the non-yeast specialist.

5) It is unclear why different experiments were done in different media. For example, if no plasmid was present YPD media was used and if plasmid was present a synthetic media was used.

We provide the information on which medium was used now consistently in all figure legends and in the materials and methods section. With respect to the use of media: in the case that plasmids were involved, cells were always grown in S-medium. YP medium was used in acute stress assays to allow for best possible growth conditions.

This causes potential reader interpretation issues in places e.g. Figure S4F/G - is the massive difference in cell survival of the Dprx1 vs Dprx1+empty plasmid due to the empty plasmid or the media? And cross-comparison with Fig 5E becomes difficult as the media used in the assay apparently changes.

We think it is a strength of our manuscript that the phenomena we report hold true across a range of conditions, for example on different media, with different carbon sources, rather than that the differences can only be observed under specific conditions. Whilst indeed with some assays we observe that media or carbon source can affect the result in terms of absolute numbers, we feel that this is to be expected and consider that what is most important is that the relative differences hold true across all conditions tested.

The medium between the indicated Figures is different: YPD (Figure **S4F**), SGal (Figure **S4G**), and SD (Figure **5E**). Thus, although cross-comparing experiments is tempting, it is not always possible, like in this case. The controls of each experiment are provided within the dataset (e.g. Figure **S4F** tests the effect of “not having matrix Prx1”(WT vs $\Delta prx1$); Figure **S4G** tests the effect of introducing matrix Prx1 in the *PRX1* deletion strain, adding the different metabolic condition “galactose”; Figure **5E** tests the effect of introducing Prx1 in the *PRX1* deletion strain, but in “glucose” and expands the dataset to the Prx1 variants C91A and P233stop.

What was the "fermentative media" used for the growth curves in Figure S2?

This was indicated (SD = glucose). In the revised manuscript, we explain the experimental conditions more extensively.

6) In the GFP-sensor assays "A steady state was measured for 10 minutes, then the sample wells were subjected to experimental treatments". Examining the figures there appears to be large differences in what is happening during this "steady state", with some being steady, but others showing significant increases during this time. How is this corrected for in calculating relative maximal changes? This for example impacts on Figure 6B.

We are not completely sure what the referee is referring to. The steady states are different (e.g. E_{GSH} is different in wild-type vs $\Delta glr1$, or after recovery from different H_2O_2 pretreatments, E_{GSH} values are different). Under certain conditions (i.e. when using the Tsa2-probe) we observe a decrease in the probe signal. This decrease results from the experimental setup and the resulting depletion of oxygen from the yeast cell pellet and has been mechanistically explained before (Morgan et al, Nature Chem Biol, 2016). In **Figure 4B**, the steady state values are all different because cells recovered from preceding treatments with different amounts of H_2O_2 .

In **Figure 5B**, we observe a slight OxD increase in the water control. We do not know the reason for this.

Specifically with respect to **Figure 6B**: The indicated value is not a relative value or a value corrected for the steady state value but the absolute maximal OxD detected in the curve (representing the highest E_{GSH}).

7) Some of the over-expressed proteins were from yeast, others were non-codon optimized human proteins (PRXD3 cloning is missing from the methods section). What effect does this have on expression levels? Many proteins were tagged (but the tag not utilized), why? Is there any evidence this tagging may modulate function? Were they equally expressed? Were they successfully targeted to the mitochondria? Where Su9 presequence was added presumably any intrinsic N-terminal targeting was removed e.g. as stated in the methods section "genes of H.sapiens PRDX5 and PRDX6 were amplified..." which implies residue 1-214 of PRDX5, but was it just the mature protein region that was cloned? If so what was the start codon used for the mature protein? Were there any linkers between the protein and the C-terminal tags? Did the C91S or P233stop codon modulate protein expression levels (they are shown as separate blots) and if so could this modify interpretation of any of the results? For the Glr1 over-expression were both the mitochondrial and cytoplasmic forms equally over-expressed?

The respective engineering experiments were performed to span the "space of H_2O_2 sensitivity", hyperoxidation capacity and susceptibility to cell death in acute stress assays. We therefore relied on matrix Grx1-roGFP2 responses to different matrix-targeted peroxiredoxin variants. We did not assess whether these proteins were expressed to similar extents (except for Prx1 variants), as this is not relevant for the following experiments that solely rely on local peroxiredoxin activity. Prx1 (C91A, P233*, WT) variants were expressed to similar extents.

We acknowledge that the information provided on these constructs in the materials and methods section was insufficient and added the missing information. We expressed PRDX3 and PRDX5 with their own targeting sequence, and added a SU9-presequence to PRDX6, AOP and Tsa1 (also indicated in **Figure 6B**).

8) The antibody against Prx1 was generated in this study. No details of how the Prx1p was purified are given nor is there any data on specificity. Hopefully larger versions of the WB shown will be included in any resubmission as supplementary figures.

We expanded the respective part of the materials and methods section. We included as supplementary figure large uncropped versions of all western blots. A

specificity analysis of the Prx1 antibody is presented in **Supplementary Figure 6E**.

9) The majority of the work is based on yeast, but the authors show some HEK293 data in Figure S1B and then comment very briefly on the large differences in [H₂O₂] requirement to elicit a response. This is not picked up elsewhere. Perhaps either more explanation should be added or the figure cut.

As suggested by this referee, we removed the data set.

10) The authors do not comment on why they observe a large effect of the water control on the apparent redox state of the matrix/cytosol e.g. figure 1G cytosol there is a large drop in signal in the Dtsa1Dtsa2 cells during the experimental time without addition of H₂O₂. This could potentially be combined with point 2 in as some supplementary text.

Indeed, in the presence of water the OxD values for the Tsa-probe (as visible in all panels in Figure 1) drops. This is due to the depletion of oxygen in the pellet that directly correlates with the amounts of H₂O₂ produced in the cells. This phenomenon is mechanistically dissected in an earlier paper introducing the Tsa-probe (Morgan et al, Nature Chem Biol 2016; also referenced in our manuscript.)

11) For the growth-curve experiments e.g. Figure S2D the authors state "cells exhibit impaired growth", but the doubling time appears to be unchanged (by eye), with the effect being in the lag-phase only. The authors should comment on this and any potential implications for the interpretation of other results.

This is indeed correct, and we now state in the manuscript that the lag phase is extended.

It is important to point out that the experiment in question is an example of chronic H₂O₂ stress, for which Prx1 is well understood to be important. It is unclear exactly why this is, speculatively it relates for example to protection of Fe-S clusters in mitochondrial proteins. Nonetheless, under these chronic conditions it is not surprising that we observe a synergistic interaction between Prx1 and Ctt1 and we feel that the data support our assertion that Ctt1 is important in preventing H₂O₂ reaching the matrix under conditions where mitochondrial redox enzymes, principally, Prx1 are impaired or absent.

Nonetheless, we do not feel that they directly impact upon our findings in terms of acute H₂O₂ stress where the presence of Prx1 is detrimental.

12) For figure 3C (1mM) it is very difficult to differentiate between the curves. For this (and other figures) the authors should consider if it is essential to use the full 0-1 scale on the y-axis.

We appreciate the comment and decided not to use the full 0–1 scale on the y-axis on all occasions.

13) The authors say "observing a significant detrimental effect of PRX1 deletion for growth in the continuous presence of H₂O₂". From Figure S4D this is true in YP media using glycerol or galactose as a carbon source, but the opposite effect is seen when using glucose as a carbon source. This needs to be commented on and any implications discussed.

On glucose-containing media ("D"), mitochondrial metabolism is low and thus production of mitochondrial reactive oxygen species. Thus, under these conditions, Prx1 might not be required and the differences between $\Delta prx1$ and wild-type during chronic H₂O₂ stress are small. This is different on galactose ("Gal") and glycerol ("G"). Here we might even expect synergistic effect of mitochondrial H₂O₂ production and external H₂O₂ leading to a requirement of Prx1 under chronic stress that has also been shown before.

14) For figure S4D what is being shown? The radius as a % compared to what? The authors should clarify if they mean "halo of growth inhibition around the disk was measured" in the methods section.

First, cells are evenly spread across the whole agar plate containing the indicated carbon source. Then, in halo assays a small filter paper disc (diameter 0.5 cm, radius 0.25 cm) is placed in the middle of an agar plate (diameter 9 cm, radius 4.5 cm). Then, the indicated concentration of reagent (H₂O₂, diamide) is applied to the filter disc. From there, it will spread likely in a non-linear fashion in a circle around the filter disc. The radius indicates how close to the filter disc (*i.e.* up to which reagent concentration) cells can still grow. 100% means no growth of cells at all (maximal radius of the halo [= no cells] of 5 cm), 0% cells grow right to the filter disc. A small scheme explains this data analysis approach (**Figure S4G**)

In a situation of chronic exposure to H₂O₂ and concomitant growth on glucose medium glucose inhibition prevents mitochondrial biogenesis and therefore any associated respiratory function and related oxidative stress, whereas in galactose and glycerol, mitochondria and the basal level of oxidative stress increase. Under these conditions, Prx1 is required to handle low amounts of continuously generated H₂O₂.

15) For figure 4G the authors may want to comment on the relative intensities of the bands between samples e.g. for 1mM why in the steady state sample the sum of bands is much more intense than for the other lanes. In addition, the unmodified sample appears to migrate slightly slower in all cases than the TCEP reduced lower band, why?

We see this often that at higher oxidant concentrations signals in immunoblotting get lower. This is also seen by others. We assume that this might be linked to differences in the efficiency of protein extraction after pretreatment of cells with oxidant and should therefore not affect the ratio between reduced and oxidized signal.

— With respect to the slightly changed migration behavior of TCEP-treated vs – untreated samples: We do not know the reasons for these differences but we do not think that this affects the interpretation of our results.

16) The legend for figure S4G is missing

We have added a figure legend.

— **17) Figure 5A, why is a modelled structure shown? The structure of Prx1p is in PDB (released October 2018).**

We have replaced the model by the actual structure.

18) Figure 5C, what is being examined, the steady state? if so this is not directly looking at hyperoxidation per se. The full assay done in figure 4G is required to eliminate between the possibilities. In addition, why in figure 4G is the ratio of the bands equal at 1mM H₂O₂, but in 5C the equivalent sample is 50% converted by 0.5mM H₂O₂? Are 4G and 5C representative WB or from a single experiment?

The reviewer is correct. The data in **Figure 5C** report steady states and thus solely accessibility to the maleimide. The experiments have been reproduced multiple times. We observed small deviations with respect to the precise transition from reduced (= accessible thiol) to oxidized (= non-accessible thiol), e.g. half-half redox state at 0.5 or 1 mM H₂O₂. **Figures 4G** and **5C** are from different biological replicates.

19) Figure S5A is a poor way of explaining the rationale behind the design. It needs more associated text. It should also be made clear that this will delete the terminal alpha-helix and 2 beta-strands. No explanation is given as to why this will make the protein more resistant to hyperoxidation.

We thank the referee for pointing this out and we now added a longer description in the Figure legend in the supplementary information.

20) There appear to be significant differences in the results shown in fig 4B (10mM) and in 5D left panel, why?

— In general, we performed all experiments with multiple biological replicates in the sets that are shown in the respective Figures. However, in the case of experiments presented in different panels, there were sometimes very long time spans before the experiments that were performed. Nonetheless, instead of combing all curves and then forming a mean and standard deviation (thereby certainly providing data that are very consistent), we decided to present our data, as they had been obtained, in the sets of biological replicates in which they were generated.

— Although these experiments exhibit some variations, for example in terms of absolute response, what we feel is most important is that the relative changes are always preserved and thus the conclusions remain the same.

21) Since the water and H₂O₂ pre-states are different in some of the samples and the effects of the water control vary is the maximal H₂O₂ response a good measure to use in Figure 6B (the response to this could possibly be combined with points 2 and 10)

There are several parameters that we could have chosen. However, we feel that the maximal OxD is the most suitable parameter. The first reason for this is that the relationship between Grx1-roGFP2 probe oxidation and GSSG level is non-linear. Changes at the higher end of the OxD scale are reflective of greater changes in GSSG than are changes in probe oxidation at the lower end of the OxD scale. Thus, we believe that maximal probe oxidation is a more reliable indicator of absolute differences in GSSG production. Moreover, we found that maximal probe OxD reliably correlates with cell death.

22) The raw data for H₂O₂ response for the empty plasmid is not shown in either 5D or S6B

We added the raw data for the empty plasmid as shown in **Figure S6B**.

Thank you for submitting a revised version of your manuscript. Please accept my apologies for the delay in communicating our decision. Your manuscript has been sent back to the three original referees. Unfortunately, one of them has not returned his/her report even after chasing.

As you will see both referee #1 and #3 find that the revised version is improved compared to the previous one and recommend the manuscript for publication. However, referee #1 requests you to revise the conclusions and in particular to tone those linking matrix glutathione oxidation and cell death.

In addition, before we can officially accept the manuscript there are a few editorial issues concerning text and figures that I need you to address.

 REFEREE REPORTS:

Referee #1:

Summary

This is a resubmission of a paper by Riemer and colleagues that conclude that mitochondrial matrix glutathione oxidation specifically promotes H₂O₂-induced cell death in *S. cerevisiae* and that, as a matrix glutathione-dependent thiol peroxidase, Prx1 regulates cell death by limiting matrix glutathione oxidation upon enzyme hyperoxidation.

General comments

Strength

As stated previously, the paper holds a number of interesting observations, some important: how the flow of H₂O₂ to the mitochondria is intercepted by cytosolic thiol peroxidases, a role of porins in facilitating H₂O₂ diffusion to the matrix, the major contributing role of Prx1 in matrix glutathione oxidation, the oxidative inactivation of Prx1 by H₂O₂, the cytosolic Ctt1 adaptation to Prx1 hyperoxidation, the protective effect of Prx1 hyperoxidation towards GSH oxidation by H₂O₂, the cell death-inducing effect of active Prx1 during H₂O₂ stress and mitigation of this effect by enzyme hyperoxidation, the correlation between GSSG accumulation and cell death.

Weakness

The demonstration that GSSG is causing cell death by H₂O₂ is not clearly answered; it can also be NADPH depletion, or an Prx intrinsic cell death signaling function.

This question is addressed in figure 7, which quests which of the NADPH depletion or the accumulation of GSSG is the cell death culprit: deletion of the glutathione reductase gene, which increases GSSG matrix levels of up to 3-fold in response to a 1 mM H₂O₂ bolus, indeed slows down growth in a Prx1-dependent and H₂O₂-independent manner (but this is growth), and decreases acute H₂O₂ tolerance at 1 and 5 mM; however, the *glr1* strain H₂O₂ tolerance is similar to WT or even better at 10 and 15 mM H₂O₂ (7C), which cannot be ignored, and if not would contradict the authors' claim. Expressing an inactivation-resistant Prx1 mutant in this strain indeed causes increased cell death at 25 mM H₂O₂, but as glutathione reductase is disabled, ongoing cycling of this Prx1 mutant should rely more on NADPH-dependent reduction by thioredoxin as an alternative reducing power, and therefore cause NADPH depletion. Overall, data clearly indicate that Prx1 is toxic in the acute stress assay at high H₂O₂ levels, presumably in part as it consumes reducing equivalents, but still cannot establish which of NADPH depletion or GSSG is causing cell death. In addition, the data confuse by the fact that Prx1 hyperoxidation occurs at 0.5-1 mM H₂O₂, but the H₂O₂ assays show differences between conditions at much higher doses of the oxidant; similarly, protection by Prx1 hyperoxidation should be seen at the low doses of H₂O₂ at which it hyper oxidizes, but the gain of function of the hyper resistant Prx1 mutant is only seen at very high doses of H₂O₂. It is also surprising that although hyperoxidation occurs at the low 0.5-1 mM H₂O₂, the Prx1-deleted strain remains much more resistant than WT at very high doses of H₂O₂ at which Prx1 is already fully hyperoxidized. To showcase their claim, authors should relate GSH oxidation and Prx1 hyperoxidation to cell death under the same conditions, at best complemented by the thioredoxin redox state, or to provide a molecular mechanism of GSSG toxicity.

Conclusion

This is an interesting paper that deserves publication, provided authors realistically address the limits of their study.

Referee #3:

This is a resubmission and general comments on the topic can be found in the original review.

The resubmission is a considerable improvement over the original, with most of the issues addressed either in full in the revised manuscript or in the point-by-point response.

It is perhaps unfortunate that the authors chose not to address point 3 from the original review. I feel that there is considerable information being overlooked by not examining the kinetics of oxidation and reduction of the probes. Rate constants for these phases may reveal some interesting differences. However, I understand why the authors chose not to proceed in this direction. I hope that the raw data will be deposited in a suitable open access databank.

I have two minor corrections:

- 1) Legend for figure S4H, it should be "filter" not "filer"
- 2) Are the significance values in figure S4L and S4M correct? For the figures I would have expected the significance to be greater

2nd Revision - authors' response

4th Jul 2019

Comments from referee #1:

We have addressed the comments from referee #1 and have now emphasized the dual role of NADPH depletion and GSSG accumulation in the discussion. The additions to the discussion are indicated using the "track changes" option.

Comments from referee #3:

We have done the two minor corrections as requested by the referee

The authors performed the requested editorial changes.

3rd Editorial Decision

8th Jul 2019

I am pleased to inform you that your manuscript has been accepted for publication in the EMBO Journal. Congratulations!

Corresponding Author Name: Jan Riemer
Journal Submitted to: The EMBO Journal
Manuscript Number: EMBOJ-2019-101552